# Human iPSC-based Modeling of Pulmonary Fibrosis Reveals p300/CBP Inhibition Suppresses Alveolar Transitional Cell State



Yusuke Tsutsui[1], Atsushi Masui[1], Satoshi Konishi[1], Taro Tsujimura[2], Mio Iwasaki[1], Takuya Yamamoto[1,2,3] & Shimpei Gotoh[1]

Idiopathic pulmonary fibrosis (IPF) is characterized by progressive scarring of lung tissue with an urgent need for effective treatments. Studies have shown that the alveolar transitional cell state (ATCS) emerges in fibrotic regions of the IPF lung. However, whether ATCS is the cause or consequence of fibrosis is controversial, and no therapeutic agents targeting the alveolar epithelial differentiation are used to treat IPF. In this study, we performed a drug screening with an in vitro pulmonary fibrosis model using fibroblast-dependent alveolar organoids derived from human induced pluripotent stem cells (iPSCs) and identified p300/CBP inhibitors as candidate therapeutic agents. Multi-omics technology revealed that ATCS induced from human iPSCs-derived alveolar organoids had a compatible profile with that reported in IPF and p300/CBP inhibitors suppressed the emergence of ATCS. Overall, these results elucidate the biological mechanisms of pulmonary fibrosis and provide a potential therapeutic target.

Idiopathic pulmonary fibrosis (IPF) is a severe disease characterized by the progressive scarring of lung tissue, which can eventually lead to respiratory failure and death[1,2]. The median survival time following an IPF diagnosis is only 2–5 years[3]. The key features of pulmonary fibrosis include excessive deposition of extracellular matrix (ECM), impaired gas exchange, and the contraction of lung tissue—driven by the contraction of activated fibroblasts[4–10]. While two FDA-approved medications have been developed for IPF, pirfenidone and nintedanib, these therapeutics only slow disease progression and are often discontinued due to significant side effects[2,11–13]. Therefore, new treatments targeting the pathogenic mechanisms of IPF are urgently needed.

Alveolar type 2 (AT2) cells are essential for maintaining alveolar homeostasis[14] and repairing damaged lung tissue[15]. This is achieved by their proliferation and differentiation into alveolar type 1 (AT1) cells, which help to restore alveolar structure and function[14]. Dysfunction of AT2 cells, including cellular senescence, has been linked to the development of IPF. Injured AT2 cells are believed to initiate pulmonary fibrosis by dysregulating the communication with fibroblasts, leading to their activation[16–20]. Consequently, a screening method for evaluating 3D cellular interactions using human AT2 cells is essential for advancing drug discovery research. However, the challenges in obtaining human AT2 cells and modeling disease pathology have limited their use in drug screening. Recent studies in mouse models of pulmonary fibrosis have revealed that alveolar transitional cell state (ATCS)—referred to as KRT8+ ADI, PATS, or DATPs—emerges as an intermediate state during the differentiation of AT2 into AT1 cells[21–23]. Additionally, genetic induction of ATCS in mice has been linked to the development of pulmonary fibrosis[6]. Conversely, reducing ATCS differentiation alleviates fibrosis in these models [24].

In patients with IPF, AT2 cells abnormally differentiate into ATCS—referred to as *KRT5*/*KRT17*$^+$ cells, aberrant basaloid cells or PATS-like cells—which accumulate in the fibrotic areas of the lungs[21,22,25–27]. Nonetheless, the molecular mechanisms that drive the emergence of these cells and the factors that regulate their abnormal differentiation remain unclear. Notably, human ATCS gene expression profiles differ from those in mice, including the expression of *TP63*, *KRT17*, and

[1]Center for iPS Cell Research and Application (CiRA), Kyoto University, Kyoto, Japan. [2]Institute for the Advanced Study of Human Biology (WPI-ASHBi), Kyoto University, Kyoto, Japan. [3]Medical-risk Avoidance Based on iPS Cells Team, RIKEN Center for Advanced Intelligence Project (AIP), Kyoto, Japan.
e-mail: gotoh.shimpei.5m@cira.kyoto-u.ac.jp

*COL1A1*[21,22,25,26]. While mouse ATCS can differentiate into AT1 cells during regeneration, human ATCS tend to persist and accumulate in fibrotic lungs, raising the question of whether this reflects a terminally arrested state or a reversible, environment-dependent transition[21,26]. This distinction highlights the need to clearly understand the specific role of human ATCS in IPF. To achieve this, establishing a system that ensures a stable supply of these cells is crucial.

We previously developed a method to generate fibroblast-dependent alveolar organoids (FD-AOs) from human pluripotent stem cells (iPSCs) in a 3D co-culture with primary human fetal lung fibroblasts (HFLFs)[28,29]. We also established a bleomycin (BLM)-induced human pulmonary fibrosis model using these FD-AOs (BLM FD-AOs)[30]. The BLM FD-AOs exhibited epithelial injury characterized by ATCS and fibroblast activation that ultimately resulted in gel contraction of the organoid matrices, which is a relevant phenotype mimicking the IPF lung, indicating disease progression and suppression.

In this study, we performed a phenotypic screening by evaluating the inhibition of gel contraction in BLM FD-AOs to identify potential therapeutic targets for pulmonary fibrosis. We identified p300/CBP inhibitors as compounds that effectively suppressed gel contraction of the FD-AO matrices. Pharmacological inhibition of p300/CBP reduced fibrotic phenotypes by regulating ATCS differentiation in both FD-AOs and mouse models of pulmonary fibrosis. Additionally, we developed a method to induce and isolate human ATCS using iPSCs, enabling us to examine the differentiation profile of these cells and the mechanisms by which p300/CBP inhibitors suppressed their differentiation. Furthermore, we explored the cell fate potential of iATCs and revealed their capacity to differentiate into AT1 and AT2 lineages, providing insights into the cellular plasticity of human ATCS associated with pulmonary fibrosis. Understanding these processes is crucial for identifying effective therapeutics for treating IPF.

## Results

### Application of BLM-induced pulmonary fibrosis model using human iPSCs to drug screening

To identify targets that suppress pulmonary fibrosis phenotypes, we conducted a phenotypic screening of small molecule compounds that inhibited gel contraction of BLM FD-AOs using human iPSCs (Fig. 1a; Supplementary Fig. 1a). The compound library comprised 264 biologically annotated compounds, including preclinical, clinical, and tool compounds (Supplementary Data 1). For contraction assessment, we utilized machine learning software (cellSense) to enhance the objectivity and throughput of the evaluation (Fig. 1b). The gel area measurements obtained through this deep learning-based evaluation showed strong consistency with those obtained through manual evaluation, as demonstrated by a high correlation ($R^2 = 0.98$). This indicates the reliability of the automated method (Fig. 1c). We assessed the gel contraction of BLM-treated FD-AOs at a compound concentration of 10 µM. The hit criteria were established at 3 standard deviations (SD) above BLM-induced contraction, with SB525334 (a TGFβ1 receptor inhibitor) serving as the positive control on each plate. Consequently, several small-molecule compounds that inhibited contraction were identified (Fig. 1d). A dose-dependent assay of the hits revealed that 19 compounds achieved 50% inhibition of gel contraction even at 1 µM (Fig. 1e). Organoids exhibiting darkened epithelial spheroids displayed increased expression of the apoptosis-related gene *NOXA*, indicating that their loss of contraction resulted from compound-induced cell death (Supplementary Fig. 1b, c). Therefore, these organoids were considered false positives and excluded from further analysis. The clustering of the evaluated compounds by signaling pathways indicated that all p300/CBP inhibitors in the library were hit compounds (8 of 8) (Fig. 1e; Supplementary Fig. 1d). Notably, CBP30 and GNE781, which are in vivo tool compounds targeting the bromodomain of p300/CBP, exhibited a potent dose-dependent inhibition of contraction in this screening[31,32]. These p300/CBP inhibitors consistently suppressed gel contraction in $SFTPC^{GFP}$ reporter iPSC (B2-3)-derived FD-AOs (Fig. 1f–h; Supplementary Fig. 1e).

### p300/CBP inhibitors suppress fibroblast activation and ATCS induction in BLM-treated FD-AOs

To clarify the molecular mechanisms responsible for the suppressed gel contraction caused by p300/CBP inhibitors, we conducted bulk RNA sequencing (RNA-seq) of isolated fibroblasts and epithelial cells from BLM/DMSO-treated FD-AOs using EpCAM antibody-based sorting (Fig. 2a). In the fibroblasts of BLM FD-AOs, treatment with p300/CBP inhibitors resulted in a reduction of *CTHRC1* and *ACTA2* expression, which are markers associated with myofibroblast differentiation and the progression of pulmonary fibrosis (Fig. 2b; Supplementary Fig. 2a)[8,9]. Pathway analysis of differentially expressed genes (DEGs) in fibroblasts from BLM FD-AOs treated with p300/CBP inhibitors showed a significant downregulation of pathways related to ECM production and gel contraction (Fig. 2c; Supplementary Fig. 2b). These findings suggest that screening for compounds with contraction-inhibitory effects may be an effective strategy for identifying potential therapeutic agents that can suppress the phenotypes associated with pulmonary fibrosis.

We next examined which cell types mediated the anti-contraction effects of the p300/CBP inhibitors. Our previous findings demonstrated that TGF-β1 stimulation induced gel contraction in a 3D fibroblast-only culture system, whereas BLM stimulation did not[30]. This indicates that the BLM-induced contraction observed in FD-AOs is mediated by epithelial cells rather than being a direct effect on fibroblasts. p300/CBP inhibitors did not inhibit gel contraction induced by TGFβ1 in a 3D fibroblast-only culture system (Supplementary Fig. 2c–e). This indicates that the suppression of fibroblast activation by p300/CBP inhibitors was likely mediated through their effects on alveolar epithelial cells. Consistently, RNA-seq analysis of alveolar epithelial cells isolated from BLM FD-AOs treated with p300/CBP inhibitors revealed a decrease in the expression of ATCS markers (Fig. 2d; Supplementary Fig. 2f). Immunostaining analysis of the active form of p300 in BLM FD-AOs showed that it was not induced in fibroblasts, but its expression was significantly increased in epithelial cells (Fig. 2e–f). Additionally, the active form of p300 was identified in SFN-positive cells, which are markers for ATCS (Fig. 2e, g). Importantly, the increases in the p300 active form and SFN expression were suppressed by p300/CBP inhibitors (Fig. 2e–g). These findings suggest that inhibiting the fibrotic phenotype in BLM FD-AOs by p300/CBP inhibitors involves a mechanism mediated by the regulation of ATCS.

In addition to the drug-induced fibrosis model using FD-AOs, we examined the efficacy of p300/CBP inhibitors in genetically induced pulmonary fibrosis models. Recent studies have demonstrated that telomere dysfunction and the resulting cellular senescence in AT2 cells contribute to the progression of pulmonary fibrosis[17]. TRF2 is a component of the shelterin complex that protects telomere length and a toxic gain-of-function variant within this complex has been reported[1,33]. Prior research has shown that cells expressing a dominant-negative variant of TRF2 (TRF2DN), which lacks its DNA-binding domain, induce cellular senescence[34,35]. To investigate this further, we generated an $SFTPC^{GFP}$ reporter iPSC line that expresses TRF2DN to model spontaneous pulmonary fibrosis. Then, we evaluated the pharmacological efficacy of p300/CBP inhibitors using this senescence-induced pulmonary fibrosis model. Using a lentiviral vector, we established an iPSC line capable of expressing TRF2DN in response to doxycycline (Supplementary Fig. 3a). In FD-AOs composed of alveolar epithelial cells derived from the iPSC line and HFLFs, we successfully induced TRF2DN in the epithelial cells in a doxycycline (DOX)-inducible manner (Supplementary Fig. 3b). Analysis revealed an increase in SA-β-Gal activity in epithelial cells but not in fibroblasts. This suggests that senescence was specifically induced in the epithelial cells of this model (Supplementary Fig. 3c). To assess the impact of


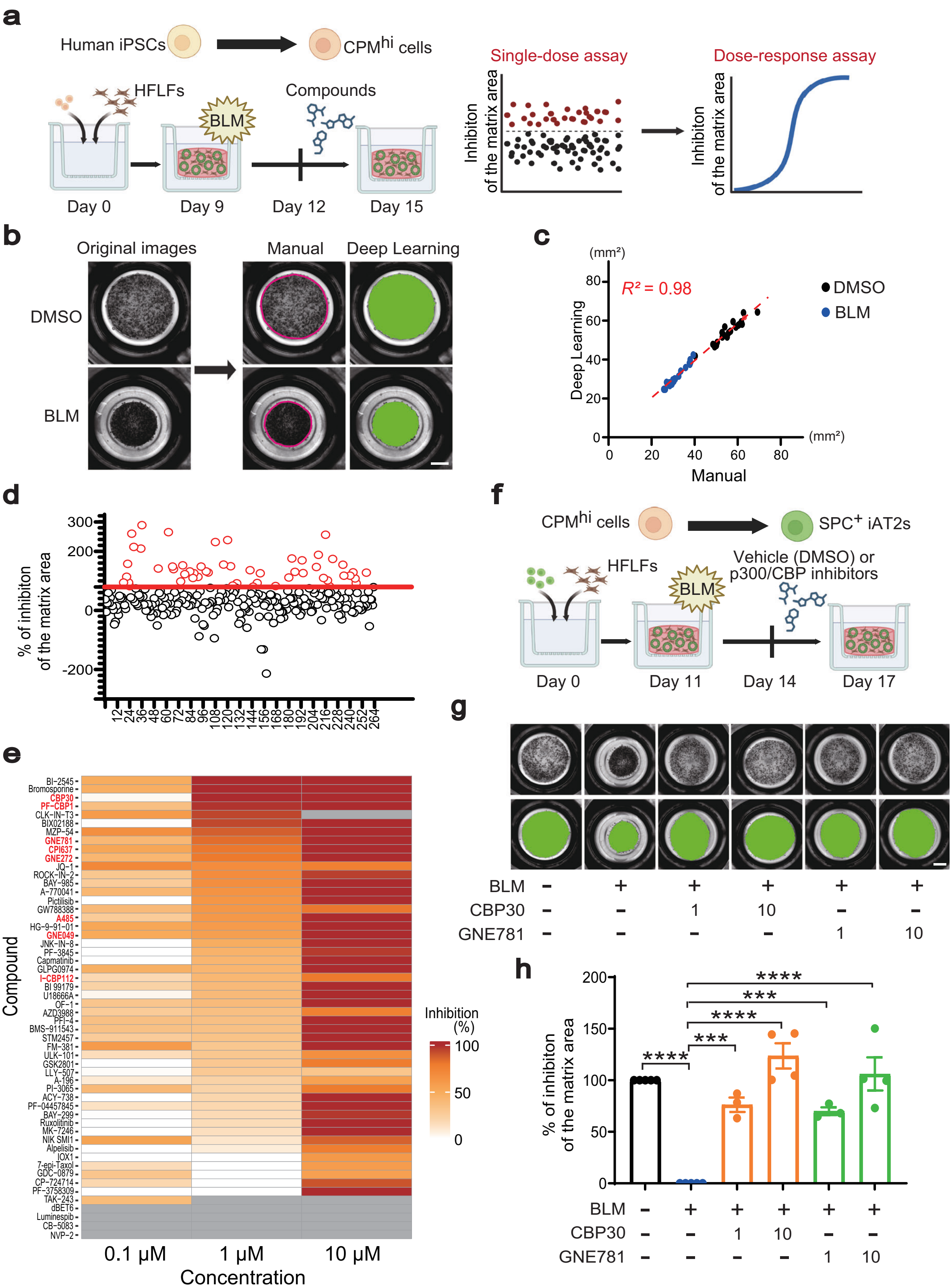


senescence in alveolar epithelial cells on the co-cultured fibroblasts, we performed RNA-seq on fibroblasts isolated from TRF2DN FD-AOs. The results showed an increased expression of fibroblast activation markers such as *CTHRC1* and *CCN2* (Supplementary Fig. 3d). Pathway analysis of DEGs from this RNA-seq indicated an enrichment of signals associated with pulmonary fibrosis (Supplementary Fig. 3e). Additionally, we observed gel contraction-related responses in TRF2DN FD-AOs, and the p300/CBP inhibitor demonstrated an inhibitory effect on this process (Supplementary Fig. 3f–g). Furthermore, p300/CBP inhibitors suppressed ATCS markers in epithelial cells while increasing the expression of AT2 markers, including *SFTPC* and *SFTPA2* (Supplementary Fig. 3h). We also evaluated the effect of the p300/CBP inhibitor on the proteomes of TRF2DN FD-AOs. Pathway analysis suggested that the p300/CBP inhibitor suppressed pathways related to

**Fig. 1 | Drug screening for inhibitors of BLM-induced gel contraction model of pulmonary fibrosis using FD-AOs. a** Schematic outline of the drug screening strategy. Created in BioRender. Tsutsui, Y. (2026) https://BioRender.com/h72b050 **b** Whole-well imaging of FD-AOs at day 17. The green areas indicate the region identified as gel using machine learning. Scale bar: 2 mm. **c** Comparison of manual and machine learning-based quantification of the gel area. The green area was recognized as gel. n = 20. **d** Dot plots showing the quantified inhibition of gel contraction in FD-AOs after treatment with 3 μg/mL BLM and 10 μM small molecules for 3 days. **e** Heatmap displaying the quantified inhibition of gel contraction in FD-AOs after treatment with 3 μg/mL BLM and 10 μM small molecules for 3 days. Each compound was evaluated at three different concentrations. The gray areas in the heatmap indicate doses at which cytotoxicity was observed. **f** Schematic outline for quantifying gel contraction in GFP⁺ iAT2-derived FD-AOs treated with BLM. Created in BioRender. Tsutsui, Y. (2026) https://BioRender.com/g85c993 **g** Whole-well imaging of GFP⁺ iAT2-derived FD-AOs treated with BLM from days 11 to 14, followed by treatment with p300/CBP inhibitors from days 14 to 17. The concentration of compounds is expressed in μM. Scale bars: 2 mm. **h** Quantification of the matrix areas. Data are presented as mean ± SEM. One-way ANOVA followed by Tukey's multiple comparisons test: $^{***}p < 0.001$, $^{****}p < 0.0001$. n = 3 (BLM + CBP30 1 μM, BLM + GNE781 1 μM), n = 4 (BLM + CBP30 10 μM, BLM + GNE781 10 μM), and *n* = 5 (DMSO, BLM) biologically independent experiments. Concentration of compounds is expressed in μM.

fibrosis and upregulated pathways associated with surfactant proteins, indicating that the inhibitor could rescue AT2 cell function impaired by senescence (Supplementary Fig. 3i–j). These findings suggest that senescence specific to alveolar epithelial cells, induced by a variant of TRF2—a component of the shelterin complex—leads to dysfunction and abnormal differentiation in AT2 cells. This, in turn, promotes fibroblast activation; importantly, the p300/CBP inhibitor effectively inhibits this process.

### CBP30, a p300/CBP inhibitor, reduces BLM-induced ATCS in vivo

To investigate whether the p300/CBP inhibitor suppresses fibrosis in vivo by inhibiting the ATCS, we administered CBP30 to mice with bleomycin-induced lung injury (Fig. 3a). The treatment resulted in a reduction in profibrotic marker expression (Fig. 3b). Additionally, CBP30 decreased the expression of ATCS markers (Fig. 3c). We observed an increase in the active form of p300 in the ATCS of the BLM-induced lung injury mice (Fig. 3d–e). Treatment with CBP30 significantly reduced the number of ATCS and myofibroblasts, which had significantly increased after BLM administration (Fig. 3f–h). These findings suggest that p300/CBP inhibition suppresses ATCS and fibrotic phenotypes in human iPSC-based models and mouse models of pulmonary fibrosis (Fig. 3i).

### Human iPSC derived-alveolar transitional cell state (iATCs) induced in the micro-patterned culture resemble bona fide ATCS in human pulmonary fibrosis

To confirm the direct regulation of p300/CBP inhibitors on the ATCS, we evaluated the effects of these inhibitors using a monoculture system of alveolar epithelial cells. Previously, we reported that the "DCIK +3i" medium induced the differentiation of iPSC-derived AT2 cells (iAT2s), whereas "Pneumacult-ALI (PAL+)" medium specifically induced functional proximal airway epithelial cells[29,36]. In a micropatterned culture plate, each medium formed apical-out alveolar and airway organoids, respectively. Notably, the PAL+ medium promoted the emergence of cells expressing ATCS markers from CPM$^{hi}$ cells[37]. The "DCIK" medium effectively induced iAT2s in FD-AOs based on established protocols for fetal AT2 cell culture[38,39]. The "3i" refers to three beneficial factors that we identified for inducing iAT2s in fibroblast-free alveolar organoids: Y27632, CHIR99021 (a canonical Wnt pathway agonist that works by inhibiting GSK-3β), and SB431542 (an inhibitor of TGF-β receptor kinase)[29]. These factors were also necessary to induce iAT2s in the micropatterned culture system (Supplementary Fig. 4a). The human ATCS develops due to impaired differentiation from AT2 cells to AT1 cells[21,22,26]. To confirm the molecular mechanisms underlying the differentiation process, we induced the differentiation of iAT2s into ATCS and AT1 cells using a micropatterned culture system, followed by single-cell multiomics analyses (Fig. 4a). For the induction of iPSC-derived AT1 cells (iAT1s), we utilized a LATS inhibitor, hereafter referred to as "DCI+LATSi," as reported previously[40,41]. Single-cell RNA sequencing (scRNA-seq) validated the differentiation of iAT2s into iATCs and iAT1s (Fig. 4b–d). We analyzed lineage scores using markers associated with ATCS, AT2 cells, and AT1 cells. Each population exhibited distinct characteristics specific to its lineage, indicating the robustness of our induction protocol in generating lineage-specific cell populations (Fig. 4e, Supplementary Table 1). We also conducted an unbiased comparison of iATCs with AT2 cells and ATCS (*KRT5⁻/KRT17⁺*cells[26]) from the published data of patients with pulmonary fibrosis, performing a comprehensive analysis based on the expression of all detected genes[26]. We found a low correlation between iATCs and AT2 cells (correlation coefficient = 0.19), whereas iATCs showed a high correlation with the gene expression profiles of ATCS from patients with pulmonary fibrosis (correlation coefficient = 0.84) (Fig. 4f; Supplementary Fig. 4b).

### Open chromatin accessibility during the differentiation of iAT2s into iATCs and iAT1s

Single-cell ATAC-seq (scATAC-seq) demonstrated that open chromatin accessibility changed distinctly under the respective induction media conditions for iATCs, iAT1s, and iAT2s (Fig. 4g–i). We transferred labels from the scRNA-seq results to the scATAC-seq data and confirmed a strong agreement between the labels, induction conditions, and scATAC-seq-based clusters (Fig. 4g–h). We extracted the cell populations labeled as iATCs, iAT1s, and iAT2s and performed an analysis of the enriched peaks within these clusters to investigate their potential regulatory roles. Enriched peaks were identified in the promoter regions of lineage marker genes specific to each cell type, providing additional epigenomic evidence for the induction of these cells (Supplementary Fig. 4c–e). Motif analysis of the peaks associated with each cluster revealed transcription factors binding to these regions. iATCs exhibited binding motifs for TP53[21], TP63[21,26], and SMAD2[42], whereas iAT2s and iAT1s displayed those for transcription factors such as FOXA2[43], NKX2-1[44], GATA6[45], and TEAD[46,47] (Supplementary Fig. 4f–h). Notably, iATCs also showed significant enrichment of transcription factors that form the AP-1 complex, including FOS, JUN, and ATF3 (Fig. 4i). These findings derived from single-cell multiomic analyses illustrate the differentiation of iAT2s into iAT1s and iATCs within the micropatterned culture system.

### iATCs isolated by CD54 activated pulmonary fibroblasts

Next, we evaluated the pharmacological effects of p300/CBP inhibitors on iATCs. Consistent with the findings from BLM- and TRF2DN-induced pulmonary fibrosis models using FD-AOs, p300/CBP inhibitors reduced the expression of markers associated with ATCS. This suggests that their beneficial effects in pulmonary fibrosis models are mediated by targeting alveolar epithelial cells (Fig. 5a). Notably, p300/CBP inhibitors did not influence the expression of AT2 cell markers, indicating that they may help to prevent the abnormal differentiation of iAT2s into iATCs (Supplementary Fig. 5a–b). To further investigate how p300/CBP inhibitors regulate the differentiation of ATCS, we isolated iATCs induced in a micropatterned culture system. scRNA-seq analysis revealed that ICAM1 (CD54) expression was notably increased in iATCs, suggesting that CD54 could be a surface marker for these cells (Fig. 5b). Immunofluorescence analysis revealed that CD54 expression was barely detectable under iAT2-inducing conditions and was weakly observed under iAT1-inducing conditions (Fig. 5c).


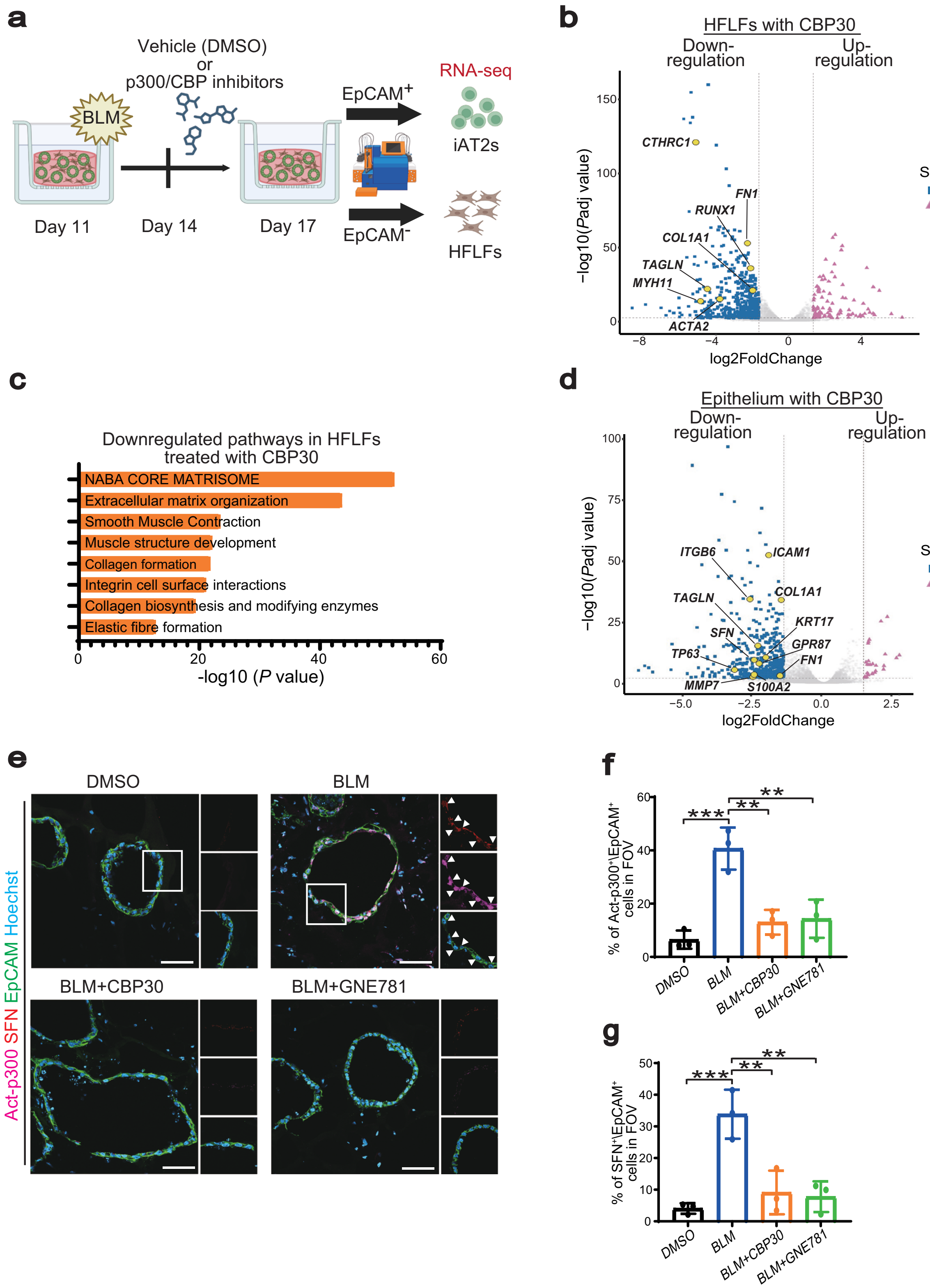


Conversely, CD54 was prominently expressed and colocalized with ATCS markers under iATCs-inducing conditions (Fig. 5c; Supplementary Fig. 5c). These results suggest that CD54 serves as a potential surface antigen for isolating iATCs, consistent with its gene expression levels (Fig. 5b; Supplementary Fig. 5d). Additionally, treatment with p300/CBP inhibitors reduced the proportion of $CD54^{+}$ cells (Fig. 5d, e). Analysis of isolated $CD54^{+}$ cells revealed increased expression of ATCS markers (Fig. 5f, g), thereby confirming that CD54 functions as a reliable marker for isolating iATCs. We previously demonstrated that primary human fetal lung fibroblasts supported the induction and maintenance of iAT2s, unlike primary normal human adult lung fibroblasts (NHLFs) that cannot induce iAT2 differentiation[29]. As idiopathic pulmonary fibrosis—a major form of pulmonary fibrosis—predominantly develops in adulthood, we investigated whether co-culture


**Fig. 2 | CBP30, a p300/CBP inhibitor, suppresses fibroblast activation and ATCS in a BLM-induced pulmonary fibrosis model of FD-AOs. a** Schematic outline for the isolation, sorting, and RNA-seq analysis conducted on the BLM-induced pulmonary fibrosis model of FD-AOs. Created in BioRender. Tsutsui, Y. (2026) https://BioRender.com/ynmoyfa **b** A volcano plot derived from differential gene expression analysis of EpCAM⁻ cells, comparing conditions with and without CBP30 (10 μM) (n = 3 biologically independent experiments). Differential expression was assessed using DESeq2 with the Wald test, and *p*-values were adjusted for multiple comparisons using the Benjamini–Hochberg method. Thresholds of $|\log_2FC| \geq 1.5$ and adjusted *p*-value (*p*adj) ≤ 0.01 are indicated by dashed lines. EpCAM⁻ cells were isolated from the BLM-induced pulmonary fibrosis model of FD-AOs. **c** Gene Ontology (GO) analysis using the top 500 downregulated genes identified by DESeq2 in EpCAM⁻ cells comparing conditions with and without CBP30 (10 μM), ranked by adjusted *p*-value. GO enrichment analysis was performed using Metascape. **d** A volcano plot derived from differential gene expression analysis of EpCAM⁺ cells, comparing conditions with or without CBP30 (10 μM) (n = 3 biologically independent experiments). Differential expression was assessed using DESeq2 with the Wald test, and p-values were adjusted for multiple comparisons using the Benjamini–Hochberg method. Thresholds of $|\log_2FC| \geq 1.5$ and *p*adj ≤ 0.01 are indicated by dashed lines. EpCAM⁺ cells were isolated from the BLM FD-AOs. **e** Representative immunofluorescence images for Act-p300, SFN, EpCAM, and nuclei (stained with Hoechst) in FD-AOs. FD-AOs were treated with p300/CBP inhibitors (10 μM) from days 14 to 17. Scale bars: 50 μm. **f** Quantification of Act-p300⁺ cells among EpCAM⁺ cells in the field of view (FOV). Data are presented as mean ± SEM. Statistical analysis was performed using one-way ANOVA followed by Tukey's test; $^{**}p < 0.01$, $^{***}p < 0.001$ (*n* = 3 biologically independent experiments). **g** Quantification of SFN⁺ cells in the FOV. Data are presented as mean ± SEM. Statistical analysis was performed using one-way ANOVA followed by Tukey's multiple comparisons test; $^{**}p < 0.01$, $^{***}p < 0.001$ (*n* = 3 biologically independent experiments).

of NHLFs with CD54⁺ iATCs promotes fibroblast activation (Fig. 5h). After co-culturing for 4 days, gene expression analysis indicated that iATCs maintained the expression of their lineage markers (Fig. 5i). Notably, the co-cultured lung fibroblasts exhibited increased expression of fibrotic markers such as *POSTN, CTHRC1*, and *COL1A1* (Fig. 5j). RNA-seq analysis revealed that alveolar fibroblast markers were downregulated, whereas fibrotic fibroblast markers were upregulated, in NHLFs co-cultured with iATCs compared with those in NHLFs cultured alone (Supplementary Fig. 6a). Pathway enrichment analyses indicated activation of fibrosis-related signaling pathways in co-cultured NHLFs (Supplementary Fig. 6b). These results suggest that ATCS directly activates alveolar fibroblasts toward a myofibroblast phenotype. NHLFs co-cultured with iATCs exhibited increased expression of several receptor genes, including *ITGAV, ITGB6, TGFBR1, FZD8*, and *NRP2* (Supplementary Fig. 6a). To identify potential mediators of epithelial–fibroblast communication, we referred to the human CellChat database[48], which indicated that TGFβ, WNT, and SEMA signaling pathways represent possible ligand–receptor interactions between iATCs and fibroblasts based on their expression patterns (Supplementary Fig. 6c, d).

### iATCs exhibit cellular plasticity toward both iAT2 and iAT1 lineages

Next, we examined the differentiation potential of iATCs. First, we evaluated whether CD54⁺ iATCs are capable of self-renewal using our previously established on-gel alveolar epithelial spheroid culture method[40] (Supplementary Fig. 7a). CCK-8 assay revealed that lung epithelial progenitor cells exhibited proliferative activity even in the iATCs induction medium, whereas CD54⁺ iATCs did not show such proliferation (Supplementary Fig. 7b). Next, we investigated whether CD54⁺ iATCs could revert to iAT2s using our previously established on-gel alveolar epithelial spheroid culture method (Fig. 6a). When CD54⁺ iATCs were cultured in our iAT2s induction medium, the expression of ATCS markers decreased, whereas the expression of AT2 lineage markers increased to levels comparable to those observed in iAT2s (Fig. 6b–d). We then evaluated whether the iAT2s derived from CD54⁺ iATCs possessed the capacity to differentiate into iAT1s using our previously developed *SFTPC*$^{GFP}$ *AGER* $^{mCherry\text{-}HiBiT}$ dual-reporter iPS cell line[40] (Supplementary Fig. 7c). Upon treatment with a LATS inhibitor, CD54⁺ iATCs-derived iAT2s showed decreased expression of AT2 markers and upregulation of AT1 markers (Supplementary Fig. 7d). These findings suggest that iATCs can be reverted into functional iAT2s capable of differentiating into iAT1s. We evaluated whether CD54⁺ iATCs could directly differentiate into iAT1 cells without reverting to iAT2s (Fig. 6e). When cultured in medium containing a LATS inhibitor, CD54⁺ iATCs showed decreased expression of ATCS markers and increased expression of AT1 lineage markers (Fig. 6f–h). AT1 marker expression levels were comparable to those observed in iAT1s derived from iAT2s using our previously established differentiation protocol (Fig. 6g). Furthermore, HiBiT luminescence was significantly increased under the iAT1-inducing condition relative to the iAT2-inducing condition, consistent with the upregulation of AT1 markers (Fig. 6i). These results suggest that CD54⁺ iATCs can be directly induced into iAT1 cells.

### H3K27ac CUT&Tag analysis revealed the regulatory mechanisms underlying iATCs differentiation by p300/CBP

To further elucidate the epigenetic mechanisms underlying iATCs differentiation, we performed CUT&Tag analysis on the isolated iATCs, iAT1s, and iAT2s. In our previous study, we demonstrated that iAT2s at the periphery of colonies induced by the micropatterned culture system can be isolated and enriched using short-term Hoechst staining (30 min)[37]. Similarly, iAT1s are predominantly localized at the colony periphery and can be enriched using the same approach (Supplementary Fig. 8a–b). By comparing these enriched populations of iAT2s (iAT2-enriched cells) and iAT1s (iAT1-enriched cells) with CD54⁺ iATCs using CUT&Tag profiling of histone modifications, we elucidated the epigenetic landscapes and associated transcription factors that shape the identity and persistence of iATCs (Fig. 7a; Supplementary Fig. 8c). For each lineage-specific gene, the peaks of activating marks were enriched in accordance with its expression (Fig. 7b; Supplementary Fig. 9). In motif analysis of H3K27 acetylation (H3K27ac), an active promoter and enhancer mark, CUT&Tag peaks in CD54⁺ iATCs identified various transcription factors, with AP-1 family transcription factors as the top-ranking motifs, agreeing with the findings from scATAC-seq (Fig. 7c). Similarly, FOX gene family motifs were most highly enriched in iAT2-enriched cells, while TEAD family motifs ranked the highest in iAT1-enriched cells (Fig. 7c). While these motifs were not entirely specific to any particular cell group and were also detected at lower ranks in the other groups, their consistent enrichment across distinct alveolar epithelial states likely reflects shared transcriptional programs associated with epithelial activation and differentiation. H3K27ac has been identified as a direct target of p300/CBP inhibition[49,50]. Immunofluorescence analysis shows a reduction in H3K27ac in alveolar epithelial cells of FD-AOs and iATCs treated with p300/CBP inhibitors (Supplementary Fig. 10a–e). Consistently, CUT&Tag analysis for H3K27ac using isolated CD54⁺ iATCs revealed peaks in the regions of ATCS marker genes (Fig. 7b; Supplementary Fig. 9). To examine the effects of p300/CBP inhibition on chromatin acetylation in iATCs, we performed CUT&Tag analysis in iATCs treated with p300/CBP inhibitors. As p300/CBP inhibition reduced the number of CD54⁺ cells, we examined an alternative isolation method using brief Hoechst staining, as that employed for iAT2- and iAT1-enriched cells. Immunofluorescence staining and flow cytometry after 30-min Hoechst labeling demonstrated that this method successfully isolated peripheral cells of each colony corresponding to CD54⁺ iATCs (Supplementary Fig. 11a, b). Using this approach, we next performed CUT&Tag analysis for H3K27ac in the presence or absence of p300/


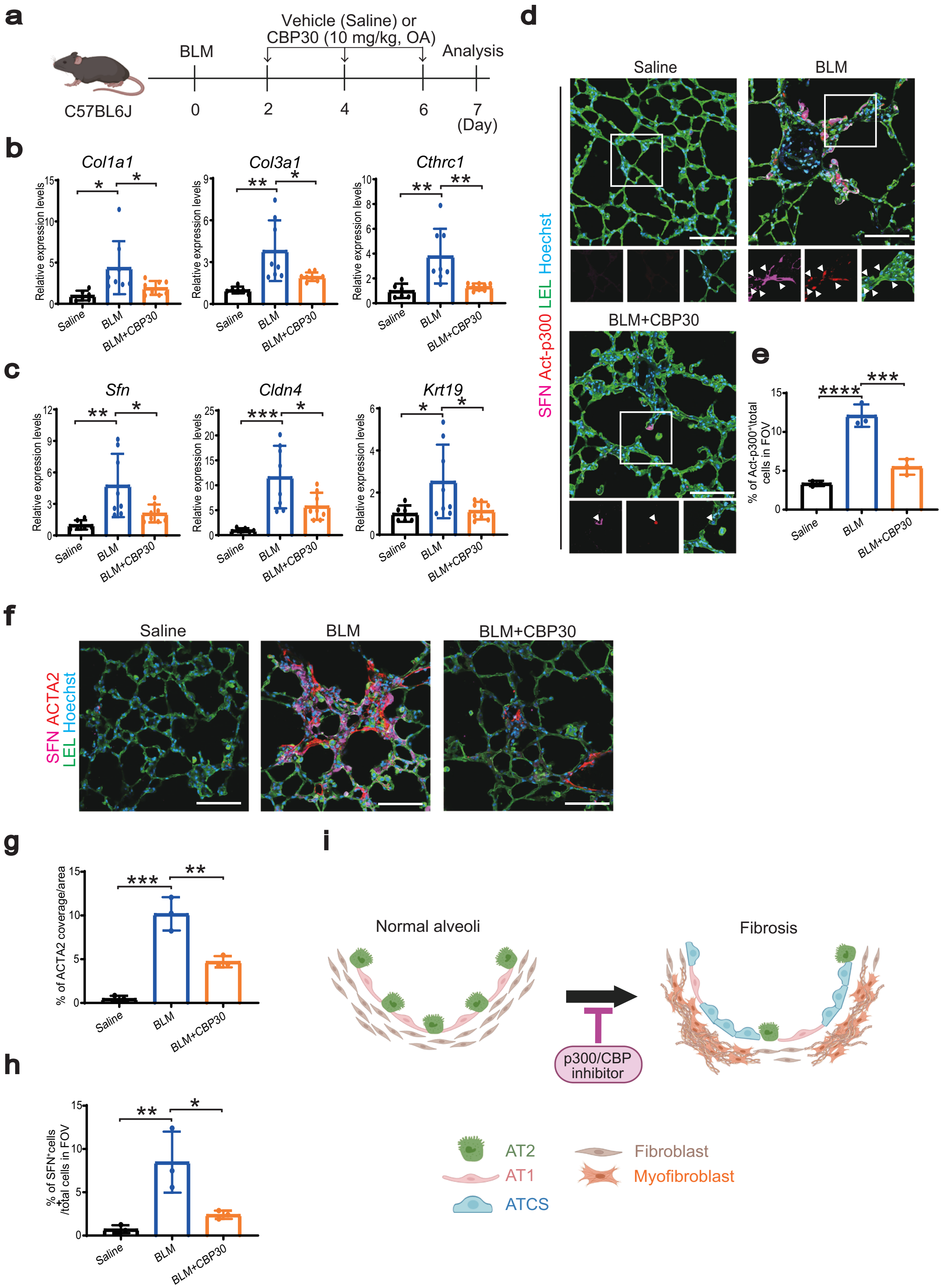


CBP inhibitors (Fig. 8a). Differential binding analysis using DiffBind revealed that H3K27ac showed a greater number of decreased peaks upon treatment with p300/CBP inhibitors (3967 and 2759 peaks with CBP30 and GNE781, respectively), whereas H3K4me3 and H3K27me3 displayed relatively few changes (Supplementary Fig. 12a). These findings suggest that p300/CBP inhibition primarily affects enhancer- and promoter-associated acetylation, rather than global chromatin states. Motif analysis of the peaks decreased by the inhibitors revealed enrichment of AP-1 motifs, consistent with the comparison between CD54$^{+}$ iATCs and iAT2s/iAT1s-enriched cells (Figs. 7c and 8b–c; Supplementary Fig. 12b). Notably, the HNF1B motif was among the top-ranked motifs in the p300/CBP inhibitor–decreased H3K27ac peaks, following widely detected motifs such as TEAD and FOX, which appeared across multiple alveolar epithelial lineages. In contrast,

**Fig. 3 | CBP30 attenuates BLM-induced ATCS differentiation in mice.** **a** Schematic outline of the mouse experiments. Created in BioRender. Tsutsui, Y. (2026) https://BioRender.com/b13d333 **b** Gene expression levels of fibrotic markers in whole lung samples under different conditions. Data are presented as mean ± SEM. Statistical analysis was performed using one-way ANOVA followed by Tukey's multiple comparisons test, with significance levels indicated as $^{*}p < 0.05$ and $^{**}p < 0.01$. $n = 6$ (Saline), n = 8 (BLM), and n = 8 (BLM + CBP30) independent mice. **c** Gene expression levels of alveolar transitional cell state markers in whole lung samples under various conditions. Data are presented as mean ± SEM. Statistical analysis was conducted using one-way ANOVA followed by Tukey's multiple comparisons test, with significance indicated as $^{*}p < 0.05$, $^{**}p < 0.01$, and $^{***}p < 0.001$. n = 6 (Saline), $n = 8$ (BLM), and $n = 8$ (BLM + CBP30) independent mice. **d** Representative immunofluorescence images showing Act-p300, SFN, nuclei (Hoechst staining), and LEL in mice with BLM-induced lung injury on day 7. Scale bar: 200 µm; inset scale bar: 10 µm. **e** Quantification of Act-p300$^{+}$ cells in the field of view (FOV). Data are presented as mean ± SEM. Statistical analysis was performed using one-way ANOVA followed by Tukey's multiple comparisons test, with significance levels of $^{***}p < 0.001$ and $^{****}p < 0.0001$ ($n = 3$ independent mice). **f** Representative immunofluorescence images showing ACTA2, SFN, nuclei (Hoechst staining), and LEL in mice with BLM-induced lung injury on day 7. Scale bar: 200 µm. **g, h** Quantification of the ratio of ACTA2 coverage per area and the number of SFN$^{+}$ cells relative to total cells. Data are presented as mean ± SEM. Statistical analysis was conducted using one-way ANOVA followed by Tukey's multiple comparisons test, with significance levels of $^{*}p < 0.05$, $^{**}p < 0.01$, and $^{***}p < 0.001$ ($n = 3$ independent mice). **i** Working model illustrating the mode of action of p300/CBP inhibitors in pulmonary fibrosis. Created in BioRender. Tsutsui, Y. (2026) https://BioRender.com/pkuir8i.

HNF1B was not highly ranked in iAT1- or iAT2-enriched cells, suggesting a potential HNF1B-specific role in the regulation of iATCs by p300/CBP (Figs. 7c and 8b–c; Supplementary Fig. 12b). Consistently, motif analysis of p300-bound regions also revealed enrichment of AP−1 and HNF1B, reinforcing their potential roles in iATCs differentiation (Supplementary Fig. 12c). To elucidate the downstream target genes and signaling pathways regulated by transcription factors AP-1 and HNF1B that cooperate with p300/CBP, we predicted their putative target genes using GREAT[51] analysis based on motifs in H3K27ac peaks reduced by p300/CBP inhibition. Among the 325 genes associated with both AP-1 and HNF1B motifs, scRNA-seq analysis identified 126 genes as being upregulated in iATCs compared with those of iAT2s and iAT1s (Figs. 4a–e and 8d–e; Supplementary Data 2). Ingenuity Pathway Analysis (IPA) of these 126 genes revealed enrichment of pathways related to epithelial stress responses and fibrogenesis, including the RHO GTPase cycle, FAK signaling, and RAF/MAPK cascade (Fig. 8f). These findings suggest that p300/CBP inhibition regulates iATCs differentiation by modulating fibrotic and cytoskeletal gene programs through AP-1- and HNF1B-dependent transcriptional control.

### p300 CUT&Tag analysis identified mechanisms underlying the fibrotic response of alveolar epithelial cells in a human pulmonary fibrosis model

To characterize the human ATCS differentiation in terms of the dynamics of p300 binding, we performed CUT&Tag analysis for p300 in FD-AOs treated with or without BLM (Fig. 9a). BLM was added at day 11, and analyses were conducted at day 14 (the timepoint when BLM washout and compound were added) and day 17 (the time point for evaluating gel contraction). Motif analysis of epithelial cells at day 14 indicated enrichment of several stress- and differentiation-related transcription factors, including p53 and HNF1B, whereas analysis at day 17 showed enrichment of SMAD2/3, HIF1A/1B, and NF-κB (p50/p52) motifs (Fig. 9a; Supplementary Data 3). Consistently, RNA-seq analysis conducted at the corresponding time point revealed these transcription factors as potential upstream regulators based on IPA, and Gene Set Enrichment Analysis further indicated activation of each signaling pathway (Supplementary Fig. 13a–b). In contrast, AP-1, TEAD, and FOX family motifs were enriched at both time points, suggesting their persistent involvement in the maintenance and progression of the fibrotic epithelial phenotype (Fig. 9a; Supplementary Data 3). These results suggest that the dynamics of transcription factors cooperating with p300 change during the transition from early epithelial injury to a pro-fibrotic state. Agreeing with these transcriptional dynamics, time-dependent alterations were also observed in peaks associated with genes involved in epithelial stress and injury responses (Fig. 9b). Moreover, motif enrichment analysis of H3K27ac peaks that increased upon BLM treatment also identified AP-1 and HNF1B motifs (Supplementary Fig. 14a–b), further supporting their involvement in p300-associated regulatory programs. In contrast, few p300 peaks were significantly reduced by BLM, and only a limited number of transcription factor motifs were detected (Supplementary Fig. 13c–d). Motif analysis of H3K27ac peaks that were significantly decreased upon BLM treatment revealed enrichment of transcription factors such as NKX2-1 and FOXA2, which are essential for alveolar epithelial development and homeostasis[43,44] (Supplementary Fig. 14c–d). Furthermore, these transcription factors were enriched in the peaks that increased upon treatment with p300/CBP inhibitors (Supplementary Fig. 14e–f). Together, these findings suggest that fibrotic stimulation suppresses the activity of epithelial lineage-defining transcription factors, thereby promoting aberrant differentiation and transition toward a pro-fibrotic epithelial state. Conversely, p300/CBP inhibition may restore proper transcriptional programs by reactivating these lineage-associated factors, counteracting epithelial dysfunction during fibrosis.

AP-1 motifs consistently ranked among the most enriched, aligning with subsequent functional experiments that revealed their critical role in driving fibrotic contraction in iATCs and BLM-treated FD-AOs (Fig. 7c; Supplementary Fig. 12b; Supplementary Data 3). To further validate the involvement of AP-1 family transcription factors in p300-mediated regulation of ATCS differentiation and transition toward a pro-fibrotic state, we examined the effects of AP-1 inhibitors (T-5224 and SR11302)[52]. Treatment with AP-1 inhibitors significantly suppressed iATCs differentiation as well as gel contraction of BLM-treated FD-AOs (Fig. 9c–d; Supplementary Fig. 15a). Among the AP-1 family members, we focused on ATF3, which is highly expressed in human ATCS and whose AT2-specific inhibition attenuates fibrosis in BLM-induced mouse models[53] (Supplementary Fig. 15b). Further, HNF1B, another transcription factor enriched in our motif analysis (Figs. 7c, 8b, c, and 9a), is expressed in mid-to-late tip cells during human lung development[54]; however, its potential involvement in ATCS differentiation, as observed in our CUT&Tag analysis, has not been previously reported. Notably, analysis of published single-cell RNA-seq datasets revealed that HNF1B expression was predominantly observed in ATCS but barely detectable in AT2 or AT1 cells in patients with pulmonary fibrosis (Supplementary Fig. 15c). To further examine the functional roles of ATF3 and HNF1B in ATCS differentiation, we performed siRNA-mediated knockdown of ATF3 and HNF1B (Fig. 9e). Genetic inhibition of ATF3 and HNF1B significantly reduced the expression of ATCS markers (Fig. 9f–g). Collectively, these results experimentally support the fact that both AP-1 family transcription factors and HNF1B participate in the p300-mediated regulation of ATCS differentiation and the transition toward a pro-fibrotic state.

In summary, our results indicate that p300, together with AP-1 and HNF1B, promote H3K27ac-mediated transcriptional regulation of iATCs, thereby linking epithelial remodeling to fibroblast activation during fibrotic change (Fig. 10).

## Discussion

In this study, we identified p300/CBP inhibitors through phenotypic screening of compounds using a gel contraction model of BLM FD-

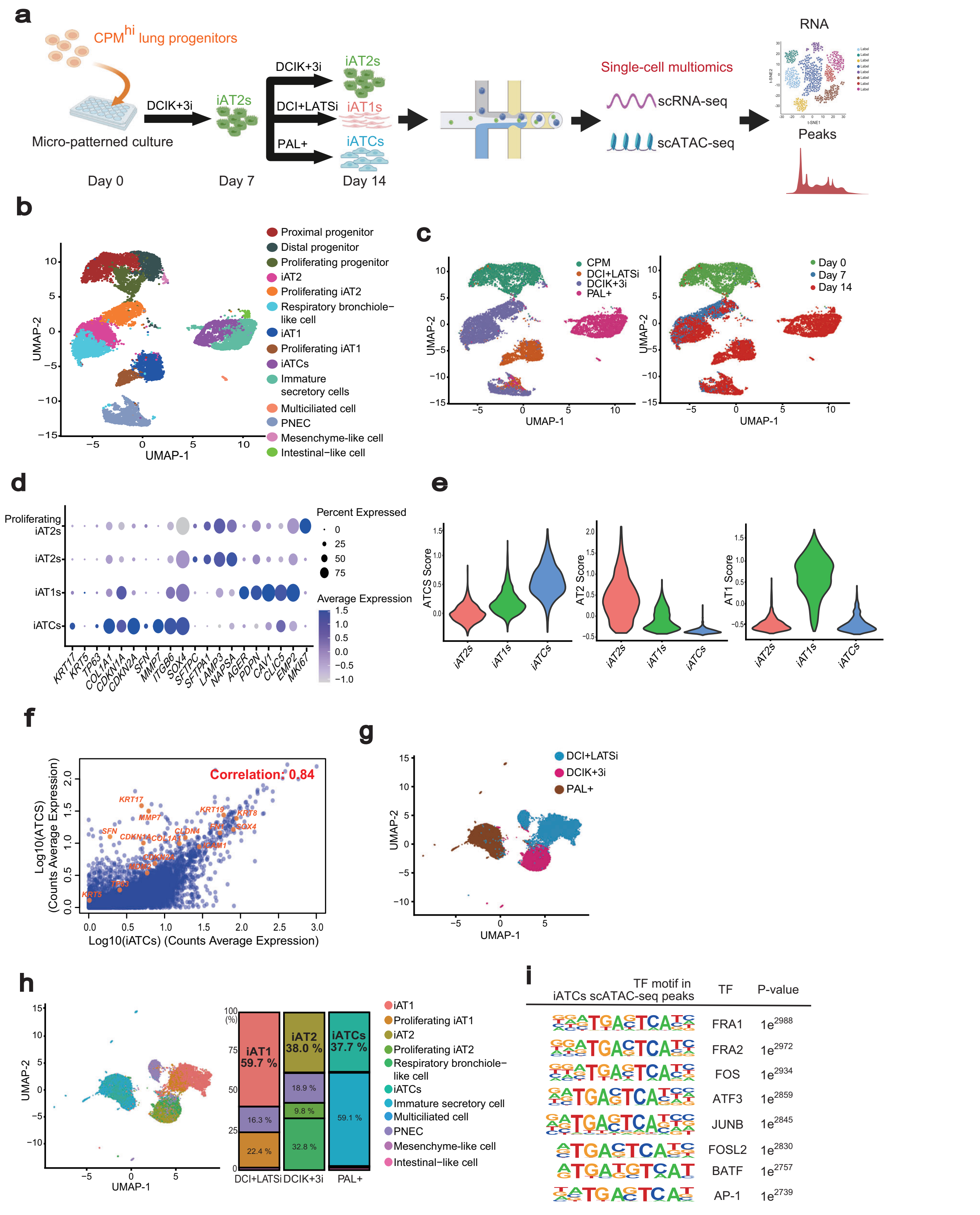


AOs. The p300/CBP inhibitors suppressed fibrotic phenotypes in both in vitro human and in vivo mouse models of pulmonary fibrosis, demonstrating the effectiveness of our human iPSC-based screening system for discovering therapeutic agents. Additionally, we demonstrated that cellular senescence could be induced in AT2 cells in a doxycyclin-dependent manner, using TRF2DN- FD-AOs. In this model, functional impairment of senescent epithelial cells led to fibroblast activation, whereas the gel contraction was induced in the senescence-driven phenotype of pulmonary fibrosis. This model served as a valuable tool for investigating the pathogenesis of spontaneously occurring pulmonary fibrosis, wherein we demonstrated that p300/CBP inhibitors mitigated the fibrosis phenotype.

Recent advances in single-cell analysis technologies have provided critical insights into the differentiation mechanisms of human

**Fig. 4 | Characterization of gene expression and open chromatin accessibility of human iPSC-derived alveolar epithelial cells in the micro-patterned culture.** **a** Schematic outline of the micro-patterned culture used for single-cell multi-omic analysis. Created in BioRender. Tsutsui, Y. (2026) https://BioRender.com/2anccku **b** UMAP visualization of major cell clusters profiled by scRNA-seq. This UMAP projection consolidates data from all samples. PNEC; Pulmonary neuroendocrine cell. **c** UMAP visualization of scRNA-seq data across different time points and culture conditions. **d** Dot plots illustrating lineage-specific markers corresponding to proliferating iAT2s, iAT2s, iAT1s, and iATCs, as shown in Fig. 4b. **e** Violin plots depicting lineage scores for ATCS, AT2, and AT1 based on defined gene sets, comparing iATCs, iAT2s, and iAT1s. **f** Scatter plots comparing the gene expression profiles of ATCS between pulmonary fibrosis patients and iPSC-derived cells. The correlation coefficient was calculated using Pearson correlation. **g** UMAP visualization of single-cell DNA accessibility profiles across each culture conditions. **h** Left; UMAP visualization of scATAC-seq data, with cell labels transferred from scRNA-seq clusters (Fig. 4B), constructed by integration with scRNA-seq data using Seurat[71] and Signac[72] packages. Right; Stacked bar chart showing the distribution of each cell type for cultured samples. The colored sections and labels show the proportion of each cell type within the total cell population for each condition. **i** AP-1 motifs identified in scATAC-seq peaks that are significantly increased in the iATCs cluster compared to the iAT2 cluster. Differential accessibility was assessed using logistic regression–based analysis implemented in Seurat (FindMarkers, test.use = “LR”), with multiple-testing correction applied (adjusted $p$-value < 0.05).

alveolar epithelial cells[54–56]. However, characterizing the precise differentiation pathway of ATCS and its targeted drug discovery remain challenging due to limited access to patient-derived samples. The present study also emphasizes that iATCs derived from iPSCs is a reliable tool to study human ATCS, displaying their potential for investigating the mechanism of differentiation. Single-cell multiomics analyses demonstrated that the micropatterning culture system enabled differentiation of iAT2s into iAT1s or iATCs, depending on the culture condition. iATCs exhibit transcriptome and ATCS marker protein expression similar to those of human ATCS reported in the lung tissues of patients with IPF. The iATCs have never been compared with ATCS derived from patients with IPF, and no other methods for generating such cells in vitro have been reported. Moreover, we developed a method to isolate iATCs using the surface marker CD54 in a micropatterned culture system, as reported during our study in parallel that transitional cell state in mice could be isolated using CD54[57]. The isolated iATCs can be used in subsequent applications, including CUT&Tag analysis to predict the efficacy of p300/CBP inhibitors on pathogenic markers of the iATCs. The isolated iATCs were co-cultured with NHLFs and used as a model to examine the contribution of ATCS to the fibroblast activation, consistently with the mouse studies[6,24,58,59]. These findings suggest that the iATCs induced through the micropatterning culture system would be a useful cellular model for studying the pathophysiology of pulmonary fibrosis and for drug discovery. This study demonstrated that isolated iATCs can be maintained in culture without self-renewal and are capable of differentiating into iAT2s and iAT1s upon appropriate medium switching. These findings further suggest that the accumulation of transitional cells in disease contexts may reflect an impaired differentiation environment rather than a terminal differentiated fate.

We observed epigenetic changes in the transcriptional regulatory regions of NKX2-1 in iATCs, leading to its reduced expression. This finding is consistent with those of previous studies in mice[44,47] and further supports the critical role of NKX2-1 in the maintenance of AT2 cells. Our scATAC-seq analysis also supported the fact that the AP-1 family of transcription factors regulates ATCS differentiation, aligning with the findings of a mouse study[57]. Furthermore, AP-1 family was identified as one of the top motifs in H3K27ac CUT&Tag analysis. Given the crosstalk between the AP-1 family and p300/CBP in transcriptional regulation, this mechanism may also be involved in ATCS differentiation[60]. Histone lysine acetylation is critical for regulating chromatin organization and transcriptional activity[61]. The histone acetyltransferases p300 and CBP function as transcriptional coactivators, influencing gene expression by acetylating specific histone residues, including H3K27ac, which enhances the recruitment of transcription factors[49,50]. Consistently, CUT&Tag analysis for H3K27ac in iATCs treated with p300/CBP inhibitors suggested the involvement of AP-1. Furthermore, treatment of the BLM-induced pulmonary fibrosis model of FD-AOs with AP-1 inhibitors suppressed the fibrotic phenotype, suggesting that AP-1 may serve as a key cofactor in p300/CBP-mediated regulation. In addition to AP-1, the present study identified HNF1B as a distinct transcription factor involved in the p300/CBP-mediated transcriptional regulation of alveolar epithelial cells. Our findings indicate that HNF1B participates in the regulation of alveolar epithelial differentiation during pulmonary fibrosis, providing a new insight into the mechanisms underlying this disease. In addition to a recent study reporting that CBP30 suppressed pulmonary fibrosis in mice[62], the present study uncovered the mechanisms by which p300/CBP inhibitors hampered the formation of ATCS. Although another previous studies showed that AT2 cells differentiated into ATCS following injury and accumulated in fibrotic regions[27], the molecular mechanisms that directly regulate those processes remained unclear. This study clarified that p300/CBP is essential in the persistence and accumulation of iATCs. Moreover, CUT&Tag analysis of p300 in BLM-treated FD-AOs suggested that p300 cooperates with several key transcription factors during the transition from early stress and injury responses to a pro-fibrotic state. These findings not only underscore the importance of p300/CBP in the initiation and progression of pulmonary fibrosis, but also provide mechanistic insight into how this coactivator network contributes to disease pathogenesis. Further analysis of the efficacy of p300/CBP inhibitors on abnormal ATCS and other pulmonary fibrosis-related characteristics will confirm their potential as candidate therapeutic agents.

# Methods

## Cell culture

SFTPC$^{GFP}$ reporter iPSC line (B2-3)[28], TRF2DN-expressing iPSC line (T67), and SFTPC$^{GFP}$ AGER$^{mCherry\text{-}HiBiT}$ dual-reporter iPSC line (B2-3-dual-42-7) were maintained in Essential 8 medium (Thermo Fisher Scientific, A1517001) without feeder cells, as described previously[28]. Human HFLFs (17.5 weeks of gestation; DV Biologics #PP002-F-1349, lot 121109 VA) were cultured in Dulbecco’s Modified Eagle’s medium (Nacalai Tesque, 08459-64) supplemented with 50 U/mL penicillin-streptomycin (Thermo Fisher Scientific, 15140122) and 10% fetal bovine serum (Sigma Aldrich, SF7524). NHLFs (aged 37 years, Male: Lonza, CC-2512, lot 20TL293907) were cultured in FGM™-2 BulletKit™ Growth Media (Lonza, CC-3132). The use of all cells, including B2-3, T67, and B2-3-dual-42-7, was exempted from ethical approval, in accordance with the guidelines of the Ethics Committee of CiRA, Graduate School and Faculty of Medicine at Kyoto University.

## Generation of TRF2DN-expressing iPSC line

To generate a DOX-inducible iPSC line expressing TRF2DN, lentiviral particles were designed and synthesized containing the mCherry reporter gene (Vector Builder). Before infection, the B2-3 iPSCs were seeded at a density of $6 \times 10^4$ cells per well in a 12-well plate coated with iMatrix-511 (Takara Bio, T311) and maintained in StemFit AK02N medium (Ajinomoto, AK02N) supplemented with penicillin-streptomycin. The following day, B2-3 cells were infected with lentiviral particles with 8 μg/mL polybrene (Nacalai Tesque, 12996-81). After adding the viral particles, the cell culture was centrifuged at 280 × $g$ for 60 min at room temperature to enhance infection efficiency. Subsequently, cells were incubated in a 37 °C $CO_2$ incubator for 6 h. The medium was then replaced with fresh StemFit AK02N with penicillin-streptomycin, and


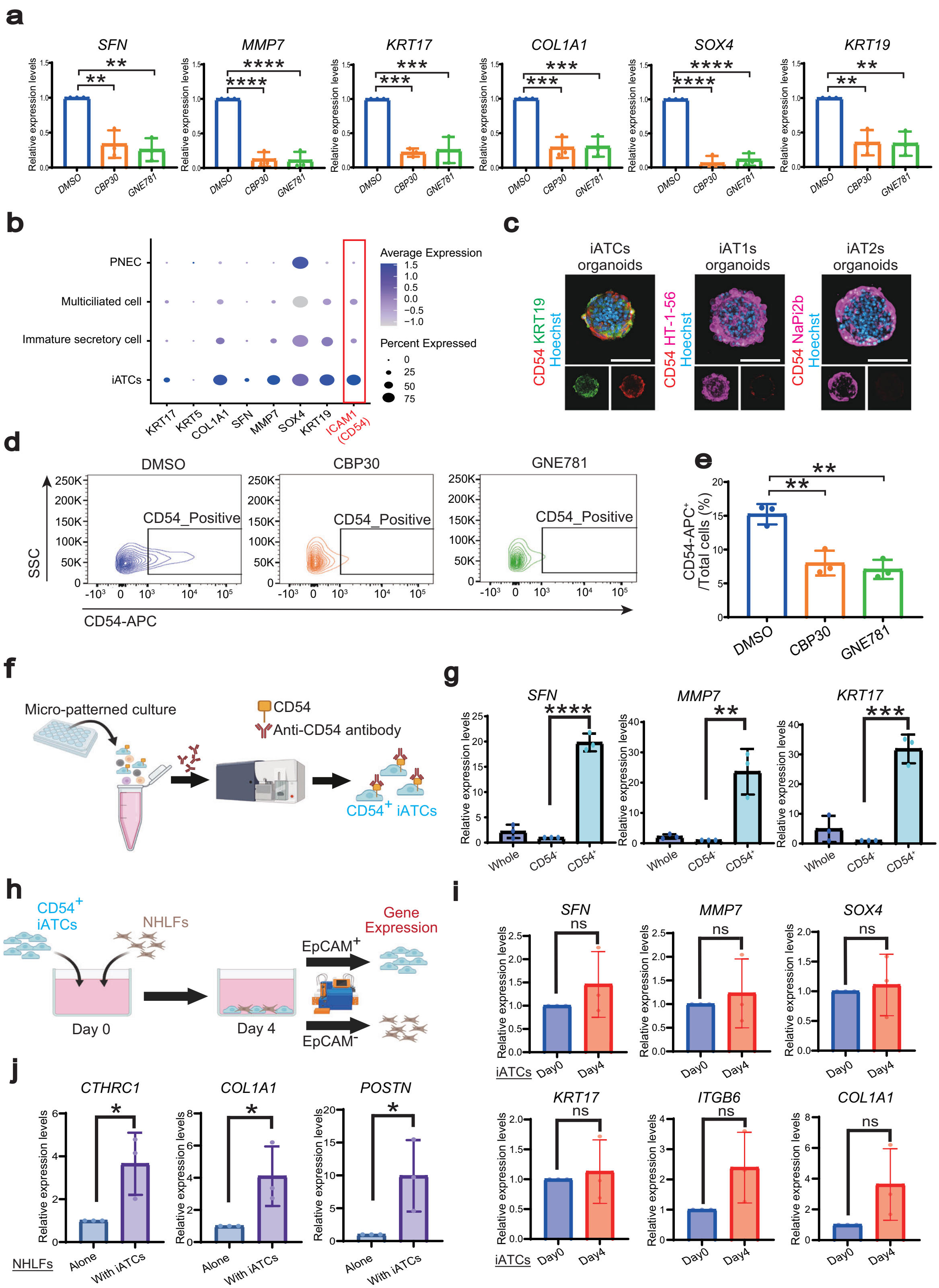


the cells were cultured for an additional 48–72 h. The mCherry-positive cells were then isolated as transduced cell population using FACS. Cells were seeded at limiting dilutions of 1500 cells per 10 cm dish in StemFit AK02N medium supplemented with 10 μM Y27632. The dishes were pre-coated with iMatrix-511, and cells were expanded in StemFit AK02N medium for 1 week. Single colonies were then manually picked using 1000 μL pipette tips under a microscope and transferred to individual wells of a 24-well plate pre-coated with iMatrix-511 and containing fresh StemFit AK02N medium. The clonal populations were expanded in culture, and the expression of TRF2DN was confirmed by the incorporated FLAG antigen using immunofluorescence microscopy and the HiBiT assay.

**Fig. 5 | CD54⁺ iATCs activated pulmonary fibroblasts. a** Gene expression of ATCS markers in the micro-patterned culture. Each well was treated with either DMSO or p300/CBP inhibitors (10 μM) from days 11 to 14. Data are presented as mean ± SEM (n = 3 biologically independent experiments). One-way ANOVA followed by Tukey's multiple comparisons test; $^{**}p < 0.01$, $^{***}p < 0.001$, $^{****}p < 0.0001$. **b** Dot plots displaying the gene expression of iATCs-specific markers in representative cell populations from the micro-patterned culture, corresponding to those shown in Fig. 4B. **c** Immunostaining of CD54 (ICAM1), lineage markers for ATCS (KRT19), AT1 cells (HT1-56), AT2 cells (NaPi2b), and nuclei (Hoechst) in micro-patterned cultures. Representative images from three biologically independent experiments with similar results are shown. Scale bar: 100 μm. **d, e** Flow cytometry analysis assessing the $CD54^+$ cell ratio in the micro-patterned culture. Each well was treated with either DMSO or p300/CBP inhibitors (10 μM) from days 11 to 14. Data are presented as mean ± SEM (n = 3 biologically independent experiments). One-way ANOVA followed by Tukey's multiple comparisons test; $^{**}p < 0.01$. **f** Schematic outline for sorting $CD54^+$ iATCs from day 14 of the micro-patterned culture. Created in BioRender. Tsutsui, Y. (2026) https://BioRender.com/v51i925 **g** Gene expression data of ATCS markers in the micro-patterned culture. Data are presented as mean ± SEM (*n* = 3 biologically independent experiments). One-way ANOVA followed by Tukey's multiple comparisons test; $^{**}p < 0.01$, $^{***}p < 0.001$, $^{****}p < 0.0001$. **h** Schematic outline of the co-culture experiment involving isolated $CD54^+$ iATCs and NHLFs. Created in BioRender. Tsutsui, Y. (2026) https://BioRender.com/nbi6c0b **i** Gene expression analysis of ATCS markers in isolated $CD54^+$ iATCs. Data are presented as mean ± SEM (*n* = 3 biologically independent experiments). One-way ANOVA followed by Tukey's multiple comparisons test; $^{**}p < 0.01$. ns; not significant ($p > 0.05$). **j** Gene expression analysis of fibroblast activation markers in NHLFs cocultured with or without iATCs. Data are presented as mean ± SEM (*n* = 3 biologically independent experiments). Unpaired two-tailed Student's t test: $^{*}p < 0.05$.

### Differentiation of iPSC into NKX2-1⁺ lung epithelial progenitor cells

The iPSCs were differentiated into lung progenitor cells, as previously described[29]. Briefly, human iPSCs were seeded onto Geltrex-coated plates (Thermo Fisher Scientific, A1413302) and cultured in RPMI-1640 medium (Nacalai Tesque, 30264-56) supplemented with Activin A (API, GF-001-050L), 1 μM CHIR99021 (Axon Medchem, CT99021), 2% B-27 supplement (Thermo Fisher Scientific, 17504-001), and 50 U/mL penicillin-streptomycin (Thermo Fisher Scientific, 15140-122). Sodium butyrate (Fujifilm Wako, 193-01522) was added at a final concentration of 0.25 mM 1 day after seeding. From days 2 to 6, cells were maintained in RPMI-1640 medium containing 100 ng/mL Activin A, 1 μM CHIR99021, 0.125 mM sodium butyrate, 2% B-27 supplement, and 50 U/mL penicillin-streptomycin. For subsequent steps, the basal medium was changed to DMEM/F12 (Thermo Fisher Scientific, 11320-033) supplemented with GlutaMAX (Thermo Fisher Scientific, 35050-061), 2% B-27 supplement, 50 U/mL penicillin-streptomycin, 0.05 mg/mL L-ascorbic acid (Fujifilm Wako, 016-04805), and 0.4 mM monothioglycerol (Fujifilm Wako, 195-15791). From days 6 to 10, cells were cultured in basal medium containing 100 ng/mL Noggin (R&D Systems, 6057-NG-01M) and 10 μM SB431542 (Fujifilm Wako, 198-16543) to induce anterior foregut endoderm (AFE) cells. From days 10 to 14, the medium was supplemented with 3 μM CHIR99021, 0.05 μM all-trans retinoic acid (Sigma-Aldrich, R2625), and 20 ng/mL BMP4 (PeproTech, 120-05ET) to generate ventralized anterior foregut endoderm (VAFE) cells. For further differentiation, from day 14 onwards, VAFE cells were maintained in basal medium supplemented with 3 μM CHIR99021, 10 ng/mL FGF10 (PeproTech, 100-26), 10 ng/mL KGF (PeproTech, AF-100-19), and 20 μM DAPT (Fujifilm Wako, 049-33583) to generate NKX2-1-positive lung progenitor cells. On day 21, the cells were labeled with anti-human CPM antibody (Fujifilm Wako, 014-27501) and $CPM^{hi}$ cells were either isolated by FACS using Alexa Fluor 647-conjugated secondary anti-mouse IgG antibody (Thermo Fisher Scientific, A31571) or by magnetic cell sorting (MACS) using anti-mouse IgG microbeads (Miltenyi Biotec, 130-048-401).

### Induction and passages of iAT2s in FD-AOs

FD-AOs were generated by co-culturing $CPM^{hi}$ lung epithelial progenitor cells with HFLFs. A total of $1.0 \times 10^4$ $CPM^{hi}$ cells were mixed with $5.0 \times 10^5$ HFLFs in 200 μL of 50% Matrigel (Corning, 354230) diluted with DCIK medium supplemented with 10 μM Y27632. The cell mixture was placed on a 12-well cell culture insert (Corning, 353180) with 1 mL of DCIK medium containing 10 μM Y27632 added to the lower chamber. DCIK medium consisted of Ham's F12 (Fujifilm Wako, 087-08335) with 50 nM dexamethasone (Sigma-Aldrich, D4902), 100 μM 8-Br-cAMP (Biolog Life Science Institute, B007), 100 μM 3-isobutyl-1-methylxanthine (Fujifilm Wako, 099-03411), 10 ng/mL KGF, 1% B-27 supplement, 0.25% bovine albumin fraction V (Thermo Fisher Scientific, 15260-037), 15 mM HEPES (Thermo Fisher Scientific, 17557-94), 0.8 mM $CaCl_2$ (Fujifilm Wako, 036- 19731), 0.1% ITS premix (Corning, 354352), and 50 U/mL penicillin-streptomycin. FD-AOs were cultured for 14 days, with the medium in the lower chamber replaced every 2–3 days. On day 14, iPSC-derived iAT2s were isolated by FACS using APC-conjugated anti-EpCAM antibody (Miltenyi Biotec, 130-113-260) to gate $EpCAM^+$/$SPC\text{-}GFP^+$ cells. Isolated iAT2 were either used for the subsequent analysis or reseeded in Matrigel (Corning, 454230) for passages. In each passage, $1.0 \times 10^4$ sorted iAT2s and $5.0 \times 10^5$ HFLFs were mixed in 200 μL of Matrigel diluted with DCIK medium and cultured on a 12-well insert. Medium was changed every 2–3 days, and passages were conducted every 2 weeks to maintain cell viability and function. When we used the TRF2DN-expressing iPSCs, FD-AOs derived from $CPM^{hi}$ cells were treated with 1 μg/mL doxycycline (Takara Bio, Z1311N) to induce TRF2DN in the epithelial cells from days 9 to 14. From days 14 to 17, the FD-AOs were cultured in dexamethasone-free DCIK medium containing 1 μg/mL doxycycline.

### Micro-patterned culture

MACS-isolated lung progenitor cells (LPCs) were seeded in 24-well micro-patterned "2.5D" culture plates (Tosoh) with multiple round cell-attachment areas, each 100 μm in diameter, precoated with 0.5 μg/cm² of iMatrix-511 at cell densities ranging from $1\times10^5$ cells/well. From days 0 to 7, LPCs were cultured in DCIK+3i medium comprised of Ham's F12 supplemented with 3 μM CHIR99021, 10 μM SB431542, and 10 μM Y27632 to promote the differentiation from LPCs to alveolar organoids (mpAOs). To maintain iAT2s, the alveolar organoids were further cultured in the same DCIK+3i medium from days 7 to 14. For the induction of iATCs, mpAOs were cultured in PneumaCult-ALI medium (Veritas, ST-05001) containing 500 nM hydrocortisone (Sigma-Aldrich, H4001), 4 μM heparin (Nacalai Tesque, 17513-54), and 10 μM Y27632 during the initial 2 days, and the same medium supplemented with 10 μM DAPT (PAL medium) for the remaining period from days 7 to 14. To differentiate iAT1 cells, mpAOs were cultured from days 7 to 14 in DCI medium without KGF, CHIR99021, SB431542, and Y27632. In addition, 10 μM LATS-IN-1 (E1061, Selleck Biotech) was added to the culture medium from days 11 to 14. The medium was replaced every 2 days throughout the entire process.

### Compound treatment in BLM-induced gel contraction model of pulmonary fibrosis using FD-AOs

Compound treatment in BLM-induced pulmonary fibrosis model using FD-AOs was performed as previously described[30]. For compound screening using FD-AOs derived from $CPM^{hi}$ cells, 3 μg/mL BLM (Nippon Kayaku) was applied from days 9 to 12. After washing out BLM with PBS on day 12, FD-AOs were cultured with each compound (Supplementary Data 1) in the medium from days 12 to 15. Similarly, FD-AOs derived from $SPC\text{-}GFP^+$ cells were treated with 3 μg/mL BLM in the lower chamber medium from days 11 to 14. On day 14, BLM was removed by washing with PBS, and the FD-AOs were cultured in dexamethasone-free DCIK medium containing the respective compounds from days 14 to 17. Compounds were purchased from MedChemExpress, and catalog numbers are provided in Supplementary

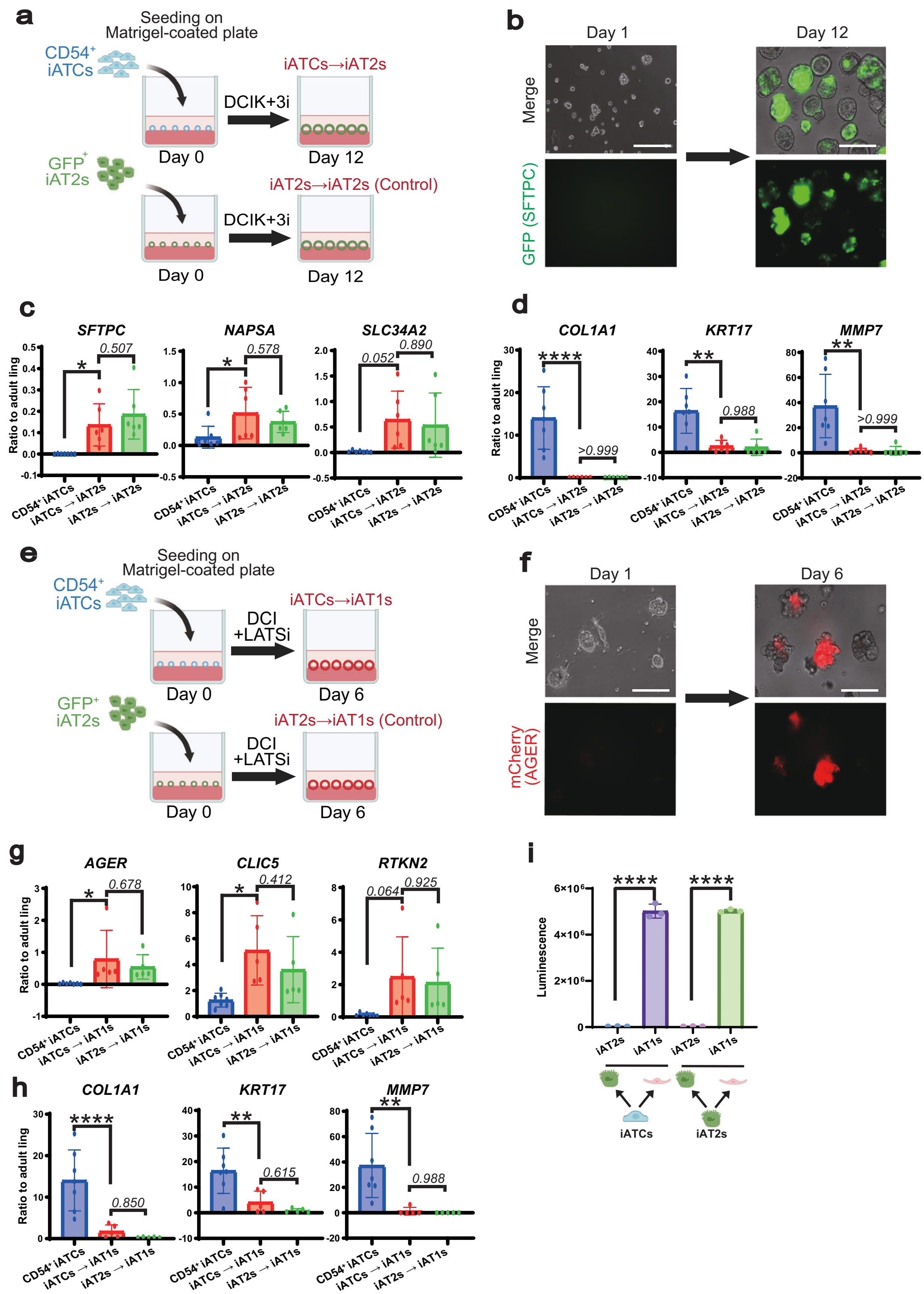


Data 1; SR11302 (S7468) and T-5224 (S8966), two AP-1 inhibitors, were purchased from Selleck Biotech.

### HFLFs-only 3D culture

HFLFs ($5.0 \times 10^5$ cells) were embedded in 200 μL of 50% Matrigel with DCIK medium containing 10 μM Y27632 and plated onto a 12-well cell culture insert (Corning, 353180). The lower chamber was filled with 1 mL of the same medium. For gel contraction assessment, 3 ng/mL active TGFβ1 (Bio-Techne, 7754-BH) and each compound were added on day 14, and the matrix areas were measured on day 17.

### Measurement of the area of the cultivation matrices of FD-AOs

The images of the gel contraction of the FD-AOs were obtained using a BZ-X710 or BZ-X810 microscope (Keyence). The gel area of FD-AOs was

**Fig. 6 | iATCs directly differentiate into iAT2s and iAT1s. a** Schematic outline for evaluating the differentiation potential of iATCs into iAT2s. Created in BioRender. Tsutsui, Y. (2026) https://BioRender.com/4a8rfce **b** Live-cell imaging of on-gel spheroids of iAT2s differentiated from iATCs. Representative images from three biologically independent experiments with similar results are shown. Scale bar: 200 µm. **c** Gene expression of AT2 markers of on-gel spheroids of $CD54^+$ iATCs-derived iAT2s and $GFP^+$ iAT2s-derived ones as a control. Data are presented as mean ± SEM. $n = 7$ ($CD54^+$ iATCs_Day0), $n = 6$ (iATCs→iAT2s), and $n = 6$ (iAT2s → iAT2s) biologically independent experiments. One-way ANOVA followed by Tukey's multiple comparisons test; $^*p < 0.05$. Numerical values indicate $p$ values. **d** Gene expression of ATCS markers of on-gel spheroids of $CD54^+$ iATCs-derived iAT2s and $GFP^+$ iAT2s-derived ones. Data are presented as mean ± SEM. n = 7 ($CD54^+$ iATCs_Day0), $n = 6$ (iATCs→iAT2s), and n = 6 (iAT2s → iAT2s) biologically independent experiments. One-way ANOVA followed by Tukey's multiple comparisons test; $^{**}p < 0.01$, $^{****}p < 0.0001$. Numerical values indicate $p$ values. **e** Schematic outline for evaluating the direct differentiation potential of iATCs into iAT1s. Created in BioRender. Tsutsui, Y. (2026) https://BioRender.com/6gskylj **f** Live-cell imaging of on-gel spheroids of iAT1s differentiated from iATCs. Representative images from three independent experiments with similar results are shown. Scale bar: 200 µm. **g** Gene expression of AT1 markers of on-gel spheroids of $CD54^+$ iATCs-derived iAT1s and $GFP^+$ iAT2s-derived iAT1s. Data are presented as mean ± SEM. $n = 7$ ($CD54^+$ iATCs_Day0), $n = 5$ (iATCs→iAT1s), and $n = 5$ (iAT2s → iAT1s) biologically independent experiments. One-way ANOVA followed by Tukey's multiple comparisons test; $^*p < 0.05$. Numerical values indicate p values. **h** Gene expression of ATCS markers of on-gel spheroids of $CD54^+$ iATCs-derived iAT1s and $GFP^+$ iAT2s-derived iAT1s. Data are presented as mean ± SEM. $n = 7$ ($CD54^+$ iATCs_Day0), $n = 5$ (iATCs→iAT1s), and $n = 5$ (iAT2s → iAT1s) biologically independent experiments. One-way ANOVA followed by Tukey's multiple comparisons test; $^{**}p < 0.01$, $^{****}p < 0.0001$. Numerical values indicate p values. **i** HiBiT luminescence under each differentiation condition. Data are presented as mean ± SEM ($n = 3$ biologically independent experiments). One-way ANOVA followed by Tukey's test; $^{****}p < 0.0001$. Schematic illustrations of cell types were created using BioRender. Created in BioRender. Tsutsui, Y. (2026) https://BioRender.com/f9kyvsy.

measured using the cellSens software (Version 4, Olympus), a deep learning-based image analysis tool employing a U-Net network architecture. The training process utilized Softmax Cross-Entropy as the loss function and Adam as the optimizer. Data augmentation included geometric transformations such as 90° rotations, mirroring, free rotation, scaling, and shearing, as well as image processing enhancements such as Random Lookup Table adjustments. Results were reported as gel area measurements ($mm^2$) for each sample.

### Isolation of epithelial cells and fibroblasts from FD-AOs

Matrigel-embedded FD-AOs were transferred into 15 mL tubes with 0.1% Trypsin-EDTA (Thermo Fisher Scientific, 25200072) and gently pipetted to dissociate the spheroids into single cells. To neutralize the trypsin, twice the volume of DMEM supplemented with 50 U/mL penicillin-streptomycin and 10% fetal bovine serum were added. The cell suspension was then centrifuged at 4 °C at 160 × $g$ for 5 min to collect the cells. The dissociated cells were stained with mouse anti-human EpCAM antibody (Santa Cruz Biotechnology, SC-66020) at a 1:100 dilution to isolate epithelial cells and fibroblasts. MACS was performed using goat anti-mouse IgG microbeads to separate $EpCAM^+$ epithelial cells and $EpCAM^-$ fibroblasts.

### RNA-seq and bioinformatics analysis

Total RNA was extracted from samples using the RNeasy Micro Kit (Qiagen) following the manufacturer's protocol. RNA integrity and concentration were assessed using the 4200 TapeStation (Agilent Technologies) for library quality control. RNA-seq libraries were prepared using the Illumina Stranded Total RNA Prep and ligated with Ribo-Zero Plus (Illumina), according to the manufacturer's instructions. Sequencing was performed on an Illumina NextSeq 2000 in paired-end mode. FASTQ files were generated using bcl2fastq v2.20, and adapter sequences along with low-quality bases were removed from raw reads using cutadapt v4.1. The trimmed reads were mapped to the human reference genome (hg38) using STAR v2.7.10a with the GENCODE annotation file (GRCh38.p13, release 32). Only uniquely mapped reads were retained for downstream analyses. Gene-level raw read counts were obtained using htseq-count v2.0.2 with the GENCODE GTF file. Differentially expressed genes (DEGs) between experimental groups were identified using DESeq2 (v1.34.0)[63] with the Wald test, and $p$-values were adjusted for multiple comparisons by the Benjamini-Hochberg method, represented as the false discovery rate (FDR). Genes with low expression (average TPM < 1 across samples) were excluded from further analyses. Gene expression was quantified as transcripts per kilobase million (TPM). Heatmaps of log2(TPM + 0.01) or DESeq2-calculated log2(fold change) values were generated using gplots. Volcano plots were visualized with the R packages ggplot2. Pathway analysis was performed using Metascape online tools and IPA software.

### Immunofluorescence analysis

Micro-patterned plates and FD-AOs were fixed with 4% paraformaldehyde (PFA) (Nacalai Tesque, 09154-56) at room temperature for 30 min. Cryosections (10 µm) were prepared after fixed specimens were immersed in 30% sucrose/PBS at 4 °C overnight, followed by embedding in OCT (Sakura Finetek, 45833). The lungs were inflated with 4% PFA and incubated at 4 °C for 4–6 h. Lung lobes were separated, washed in PBS, and incubated overnight in 30% sucrose (Fujifilm Wako, 196-00015) at 4 °C. Lung lobe samples were subsequently incubated in 1:1 30% sucrose: OCT for 1 h followed by embedding in OCT blocks and cryosectioning (10 µm). For frozen lung tissue sections, antigen retrieval was performed by incubating the sections with Antigen unmasking solution pH 6.0 (Vector Laboratories, H-3300) at 98 °C for 10 min. Cryosections were permeabilized in PBS containing 0.2% Triton X-100 and incubated in a blocking buffer of PBS containing 5% normal donkey serum (EMD-Millipore, 566460) and 1% BSA (Sigma, A9647). Samples were incubated overnight at 4 °C with primary antibodies diluted in the blocking buffer. Primary antibodies used in this study included GFP (1:500, Aves Labs, GFP-1020), SFN (1:200, Abcam, ab77187), act-p300 (1:200, biorbyt, ORB6262), EpCAM (1:200, Santa Cruz Biotechnology, sc-66020), KRT19 (1:200, Merck, MABT913), KRT17 (1:100, Abcam, ab109725), COL1A1 (1:200, Abcam, ab138492), CD54 (1:200, BioLegend, 353102), CD54 (1:200, Atlas, HPA002126), HT1-56 (1:200, Terrace Biotech, TB29AHT1-56), NaPi2b (1:200, kindly provided by Dr. Gerd Ritter (MX35)), AGER (R&D systems, AF1145), H3K27ac (1:200, Cell Signaling technology, 8173), FLAG (1:500, Cell Signaling technology, 14793), alpha smooth muscle Actin (1:100, Abcam ab5694), and Fluorescin (1:1000, Vector Laboratories, FL-1171). Following primary antibody incubation, the samples were washed in the PBS thrice and then stained with secondary antibodies at room temperature for 1 h. The secondary antibodies used in this study were Anti-rabbit IgG Alexa Fluor 546 (Thermo Fisher Scientific, A10040), Anti-rabbit IgG Alexa Fluor 647 (Thermo Fisher Scientific, A31573), Anti-mouse IgG Alexa Fluor 488 (Thermo Fisher Scientific, A21202), Anti-mouse IgG Alexa Fluor 546 (Thermo Fisher Scientific, A10040), Anti-mouse IgG Alexa Fluor 647 (Thermo Fisher Scientific, A31571), Anti-chicken IgY Alexa Fluor 488 (Thermo Fisher Scientific, A21202), Anti-goat IgG Alexa Fluor 647 (Thermo Fisher Scientific, A21447), and donkey Anti-rat IgG H&L Alexa Fluor 488 (Abcam, ab150153). All secondary antibodies were used at a concentration of 1:500. Nuclei were counterstained with Hoechst-33342 (1:1000, Dojindo, H342). Stained sections were embedded with ProLong Gold Antifade Mountant (Thermo Fisher Scientific) to preserve fluorescence signals. All samples were imaged using an LSM900 (ZEISS).

### Measurement of SA-β-gal activity

SA-β-gal activity was measured with a 96-Well Cellular Senescence Assay (Cell Biolabs, CBA-231), according to the manufacturer's


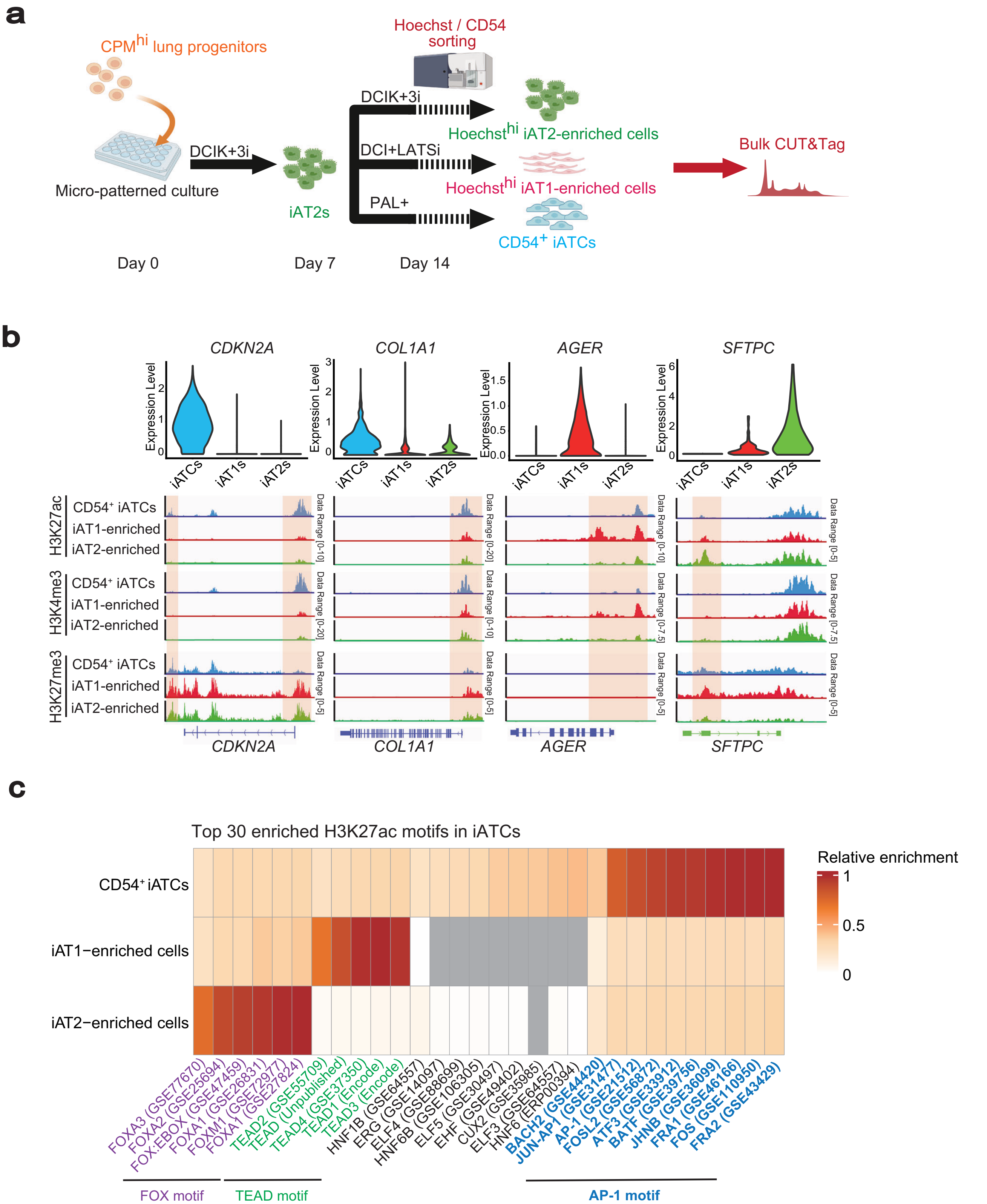


**Fig. 7 | CUT&Tag analysis of H3K27ac, H3K4me3, and H3K27me3 in CD54⁺ iATCs. a** Schematic outline of the micro-patterned culture used for CUT&Tag analysis of CD54⁺ iATCs, Hoechst$^{hi}$ iAT1-enriched cells, and Hoechst$^{hi}$ iAT2-enriched cells induced in the micro-patterned system. Created in BioRender. Tsutsui, Y. (2026) https://BioRender.com/s2t4myv **b** Violin plots of scRNA-seq expression from Fig. 4B (top) and their corresponding CUT&Tag tracks visualized in the Integrative Genomics Viewer (IGV) (bottom) for the genes *CDKN1A*, *COL1A1*, *SFTPC*, and *AGER*. **c** Heatmap showing the top 30 transcription factor motifs enriched in H3K27ac peaks of CD54⁺ iATCs compared with iAT1- and iAT2-enriched cells. Significantly increased peaks were identified using DiffBind. Motif enrichment was assessed using HOMER, and the color scale indicates relative motif enrichment, calculated by scaling the $-\log_{10}(P)$ values within each cell type so that the minimum value is set to 0 and the maximum value is set to 1 by dividing each value by the maximum (0 = lowest, 1 = highest). Gray tiles indicate motifs not detected in the respective cell type.

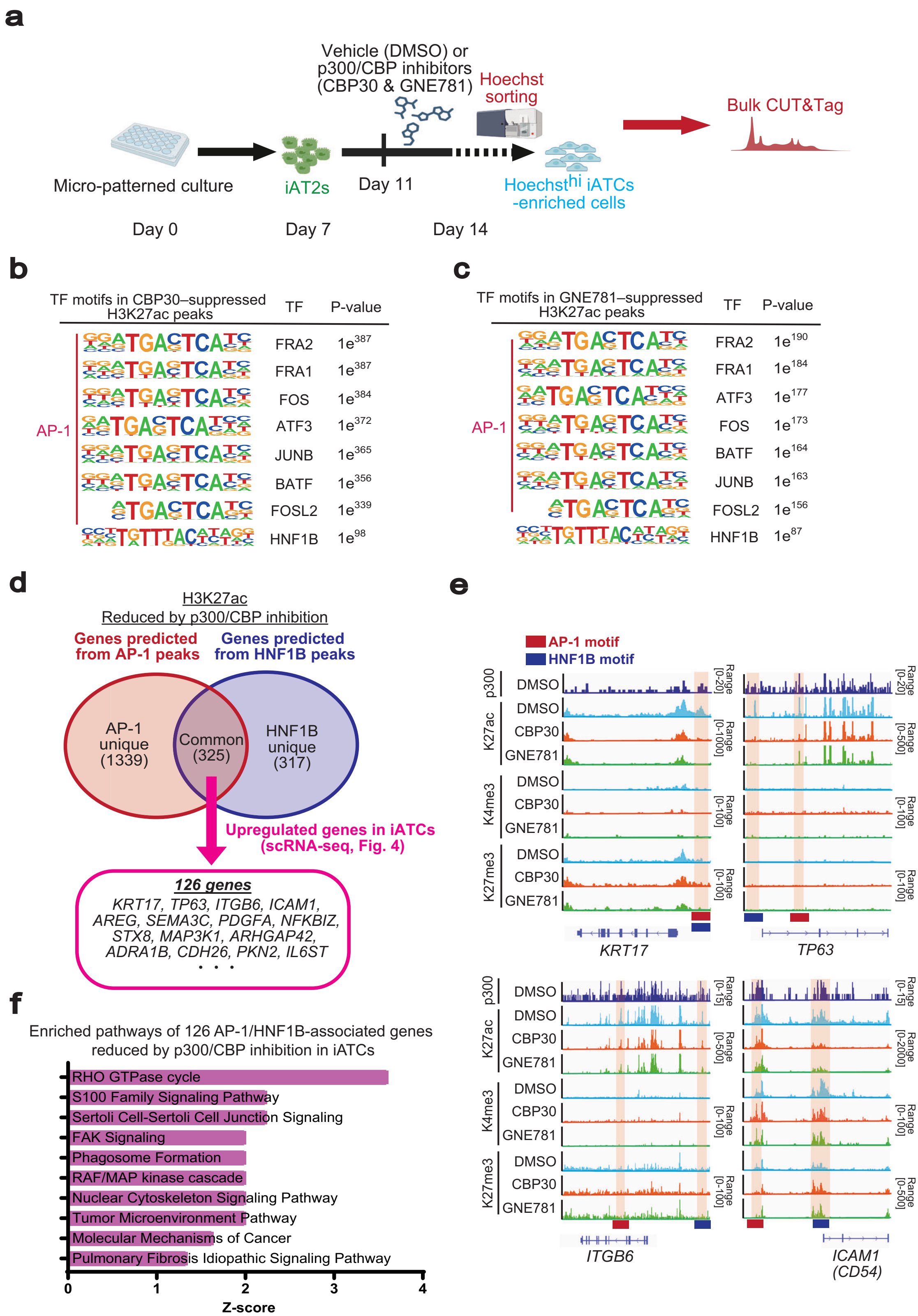


protocol. Each value of SA-β-gal activity was corrected by the protein concentration measured with a BCA protein assay kit (Thermo Fisher Scientific, 23227).

## Proteomic analysis

FD-AOs were lysed with PTS buffer[64] [12 mM sodium deoxycholate (SDC, 190-08313), 12 mM sodium lauroyl sarcosinate (SLS, 192-10382), and 100 mM Tris-HCl (pH 9.0)] containing a protease inhibitor mixture (Sigma-Aldrich, P8340). As previously described, the lysates were used as protein samples and processed using a modified protein aggregation capture method[65,66]. A total of 200 ng of desalted peptides were loaded and separated on an Aurora column (250 mm length, 75 μm i.d., IonOpticks) using a nanoElute2 (Bruker) for subsequent analysis by timsTOF Pro2 system (Bruker). The mobile phases were composed of

**Fig. 8 | Evaluation of epigenetic changes induced by p300/CBP inhibitors in CD54⁺ iATCs by CUT&Tag analysis. a** Schematic outline of CUT&Tag analysis in p300/CBP inhibitors–treated iATCs. Created in BioRender. Tsutsui, Y. (2026) https://BioRender.com/owocwcf **b-c** Representative enriched transcription factor motifs in H3K27ac peaks significantly reduced (FDR < 0.05) by CBP30 (**b**) or GNE781 (**c**) treatment. Significantly reduced peaks were identified using DiffBind, and motif enrichment analysis was performed using HOMER. **d** Identification of genes potentially co-regulated by AP-1 and HNF1B under p300/CBP inhibition. Genes associated with H3K27ac peaks reduced by p300/CBP inhibition were predicted using the GREAT program based on AP-1 and HNF1B motifs. The Venn diagram shows the overlap between AP-1– and HNF1B–associated genes. Among 325 shared genes, 126 genes were upregulated in iATCs compared with those of iAT2s and iAT1s (Fig. 4). **e** Representative CUT&Tag tracks in IGV with or without p300/CBP inhibitors. Signal intensities were normalized to *E. coli* DNA[74]. Red and blue bars indicate genomic regions containing AP-1 and HNF1B binding motifs identified by HOMER motif analysis. **f** Pathway analysis of genes co-regulated by AP-1 and HNF1B under p300/CBP inhibition. Genes predicted from AP-1 and HNF1B motifs in H3K27ac peaks reduced by p300/CBP inhibition and upregulated in iATCs (126 genes) were analyzed using IPA.

0.1% formic acid (solution A) and 0.1% formic acid in acetonitrile (solution B). A flow rate of 400 nL/min of 2–17% solution B for 60 min, 17–25% solution B for 30 min, 25–37% solution B for 10 min, 37–80% solution B for 30 s, and 80% solution B for 8.5 min was used (100 min in total). The applied spray voltage was 1500 V, and the interface heater temperature was 180 °C. To obtain MS and MS/MS spectra, the Parallel Accumulation Serial Fragmentation (PASEF) acquisition method with data-independent acquisition (DIA) mode was used (diaPASEF)[67]. For diaPASEF settings, 1.03 s per one cycle with precursor ion scan and 11 times diaPASEF scans were conducted with the MS/MS isolation width of 15 m/z, precursor ion ranges of 340–1105 m/z, ion mobility ranges of 0.75–1.32 V· s· cm^(−2). The obtained DIA data were searched by DIA-NN (v1.8.2 beta27)[68] against selected human entries of UniProt/Swiss-Prot release 2024_02 with the carbamidomethylation of cysteine as the fixed modification and protein N-terminal acetylation and methionine oxidation as the variable modification. For the other DIA-NN parameters, Trypsin/P protease, one missed cleavage, peptide length range of 7–30, precursor m/z range of 300–1800, precursor charge range of 1-4, fragment ion m/z range of 200-1800, and 1% precursor FDR were used. The values "PG.MaxLFQ" obtained from the results were used as representative protein area values to compare protein expression levels using edgeR (v3.20)[69]. The mass spectrometry data were deposited in the ProteomeXchange Consortium via jPOSTrepo[70] (https://repository.jpostdb.org/) with the dataset identifier JPST003570 (PXD060050).

### Flow cytometry
Cell suspensions were prepared using a flow cytometry buffer consisting of PBS with 1% BSA and 10 µM Y27632. The cells were incubated in Accutase (Innovative Cell Technologies, ICT-AT104-500-500) at 37 °C for 20 min and gently detached by pipetting. For staining, the cells were incubated with APC-conjugated anti-human CD54 antibody (BioLegend, 353112) for 20 min at 4 °C. Isolation of the Hoechst$^{high}$ cells was performed as previously described[37]. Briefly, cells cultured on micro-patterned plates were first stained with Hoechst-33342 for 30 min at 37 °C, followed by incubation in Accutase at 37 °C for 20 min and dissociation by gentle pipetting. Hoechst-high populations were separated based on Hoechst fluorescence intensity using FACS.

### Co-culture of iATCs and NHLFs
To co-culture iATCs isolated from micro-patterned plates with NHLFs, 12-well plates (Corning, 3513) were coated with 50 µg/mL fibronectin (Gibco, 33016-01) and incubated overnight. On the following day, the fibronectin solution was aspirated, and the wells were washed with PBS. After washing, the wells were dried for at least 30 min. Once fully dried, $3.0 \times 10^5$ iATCs and $1.5 \times 10^4$ NHLFs were suspended in PAL medium without DAPT and seeded into the prepared wells, followed by incubation in a $CO_2$ incubator. As a control, NHLFs were also seeded into wells without iATCs. The next day, the medium was changed to PAL medium without DAPT and Y27632. After an additional 3 days of culture, cells were detached using 0.1% Trypsin-EDTA, and iATCs and NHLFs were isolated following the same protocol as described in the "Isolation of Epithelial Cells from FD-AOs".

### On-gel culture
Each well of a 96-well plate was coated with 100 µL of Growth Factor–Reduced Matrigel (Corning) at least 30 min before cell seeding, as described previously[40]. For iAT2 induction, $3 \times 10^4$ CD54⁺ iATCs or GFP⁺ iAT2s were suspended in 100 µL of DCIK + 3i medium and seeded into the coated wells, followed by incubation at 37 °C in a humidified $CO_2$ incubator. The medium was replaced every 3 days. For direct iAT1s induction, $1 \times 10^5$ CD54⁺ iATCs or GFP⁺ iAT2s were suspended in 100 µL of DCI + LATSi medium supplemented with Y-27632 (10 µM) and seeded onto the coated wells. On the following day, the medium was replaced with Y-27632–free DCI + LATSi, and the culture medium was subsequently changed every 3 days.

### HiBiT assay
For iAT1s and iAT2s derived from the SFTPC$^{GFP}$ AGER$^{mCherry\text{-}HiBiT}$ dual-reporter iPSC line, luminescence was measured using the Nano-Glo HiBiT Lytic Detection System (Promega) according to the manufacturer's instructions. Luminescence intensity was measured using an EnVision 2104 Multilabel Reader (PerkinElmer).

### mRNA expression analysis
Total RNA was extracted using the RNeasy Micro Kit (QIAGEN, 74004) according to the manufacturer's protocol. ReverTra Ace qPCR RT Master Mix with gDNA Remover (Toyobo, FSQ301) was used for the reverse transcription of RNA. Quantification was performed using THUNDERBIRD Probe qPCR Mix (Toyobo, QPS101) or Power SYBR Green Master Mix (Thermo Fisher Scientific, 4368706). The primers used are listed in Supplementary Table 2 and 3. Gene expression was normalized to that of 18S rRNA. The expression of each gene was compared with that of the human adult lung five donor pool (BioChain, R1234152-P, lot A811037) or each control.

### Single-cell RNA-seq (scRNA-seq)
Single-cell RNA libraries for lung epithelial cells and LPs were prepared using a 10X Genomics Chromium device, according to the manufacturer's protocols specified in the Single Cell 3′ Reagent Kits v3.1. The 10X Genomics Cell Ranger pipeline (version 7.1.0) was used to perform sample demultiplexing, alignment to the hg38 human reference genome (refdata-gex-GRCh38-2020-A from 10X Genomics), and the reporter sequences, barcode/UMI processing, and gene counting for each cell. In the quality control (QC) of sequencing data, dead cells or outliers were identified and excluded based on two criteria: cells with fewer than 200 detected genes and cells with a high percentage of counts mapped to mitochondrial genes, with a threshold set at 15%. We used the Seurat package (v3.2.3)[71] of the R software for data analysis and visualization.

### Single-cell ATAC-seq (scATAC-seq)
scATAC-seq libraries were prepared using DNBelab C Series High-throughput Single-cell ATAC Library Preparation Set (MGI, 940-000793-00), following the manufacturer's protocol. After making the nuclei suspension, we subjected 100,000 nuclei to transposition reaction and 20,000 nuclei to droplet formation for each sample using DNBelab C-TaiM 4RS (MGI, 900-000637-00). PCR was first performed in droplets for 10 cycles. Then, after the subsequent demulsification

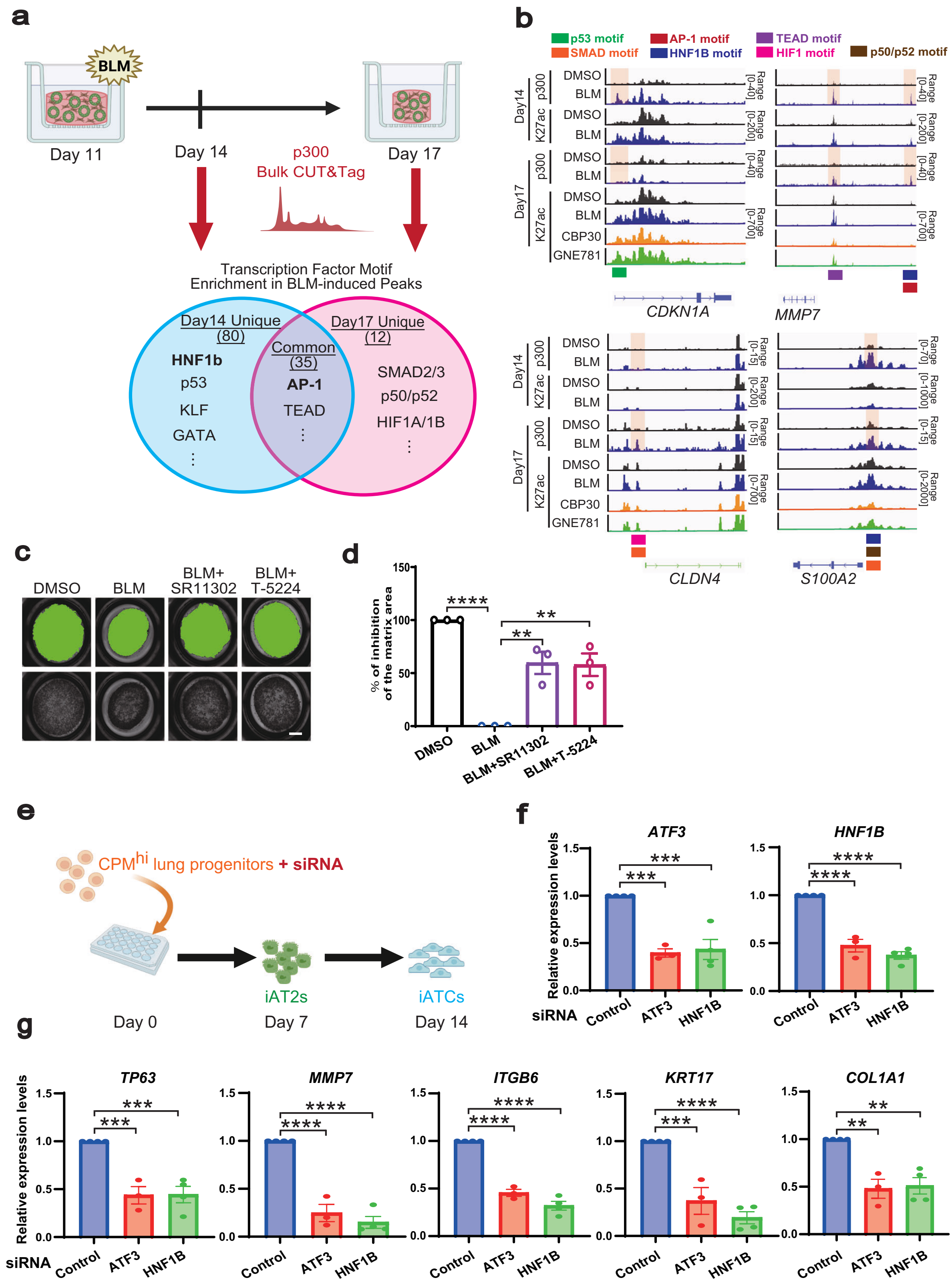


and bead purification, PCR amplification was performed with scATAC Barcode primers for 12 cycles. Then, we converted the generated libraries to DNB through single-stranded circularization and rolling amplification. DNB was loaded onto T7 flowcells with the MGIDL-T7RS loader and then sequenced with the DNBSEQ-T7RS sequencer (MGI). The sequenced read lengths were 115 cycles (1–6 bp, 17–22 bp, and 33–65 bp were set for dark reaction) for Read1, 69 cycles (1–19 bp were set for dark reaction) for Read2, and 10 cycles for the index. The generated fastq files were processed with dnbc4tools (v2.1.2). We aligned the reads to the human genome hg38 assembly after creating the genome index file together with the GENCODE v32 annotation GTF file. Then, the scATAC-seq data were processed using the Seurat and Signac (v1.13.0)[72] R packages. The sequencing data, including peak, barcode, and matrix files, were loaded, and the matrix was filtered

**Fig. 9 | p300 CUT&Tag analysis and functional assays for transcription factors mediating the fibrotic response in alveolar epithelial cells. a** Schematic outline of p300 CUT&Tag analysis in BLM-treated FD-AOs and representative transcription factor motifs enriched in p300 peaks significantly increased (adjusted *p* value < 0.05) upon BLM treatment. Created in BioRender. Tsutsui, Y. (2026) https://BioRender.com/tzol3zv. Significantly increased peaks were identified using DiffBind. Motif enrichment was assessed using HOMER. **b** Representative p300 and H3K27ac CUT&Tag tracks visualized in IGV at injury-responsive epithelial genes (*CDKN1A*, *MMP7*, *CLDN4*, and *S100A2*) in FD-AOs treated with DMSO or BLM. Signal intensities were normalized to *E. coli* DNA[74]. Colored bars indicate genomic regions containing transcription factor motifs of interest. **c** Whole-well imaging of FD-AOs treated with BLM from days 11 to 14, followed by treatment with AP-1 inhibitors (SR11302, 10 μM; T-5224, 40 μM) from days 14 to 17. Scale bars: 2 mm. **d** Quantification of the matrix areas. Data are presented as mean ± SEM. One-way ANOVA followed by Tukey's multiple comparisons test: $^{**}p < 0.01$, $^{****}p < 0.0001$ (*n* = 3 biologically independent experiments). **e** Schematic outline of siRNA-based gene silencing in the micro-patterned culture. Created in BioRender. Tsutsui, Y. (2026) https://BioRender.com/35r9mk4 **f** Gene expression data of ATF3 and HNF1B in the micro-patterned culture. Data are presented as mean ± SEM. n = 4 (siCont), *n* = 3 (siATF3), and *n* = 4 (siHNF1B) biologically independent experiments. One-way ANOVA followed by Tukey's multiple comparisons test; $^{***}p < 0.001$, $^{****}p < 0.0001$. **g** Gene expression data of ATCS markers. Data are presented as mean ± SEM. *n* = 4 (siCont), *n* = 3 (siATF3), and *n* = 4 (siHNF1B) biologically independent experiments. One-way ANOVA followed by Tukey's test; $^{**}p < 0.01$, $^{***}p < 0.001$, $^{****}p < 0.0001$.

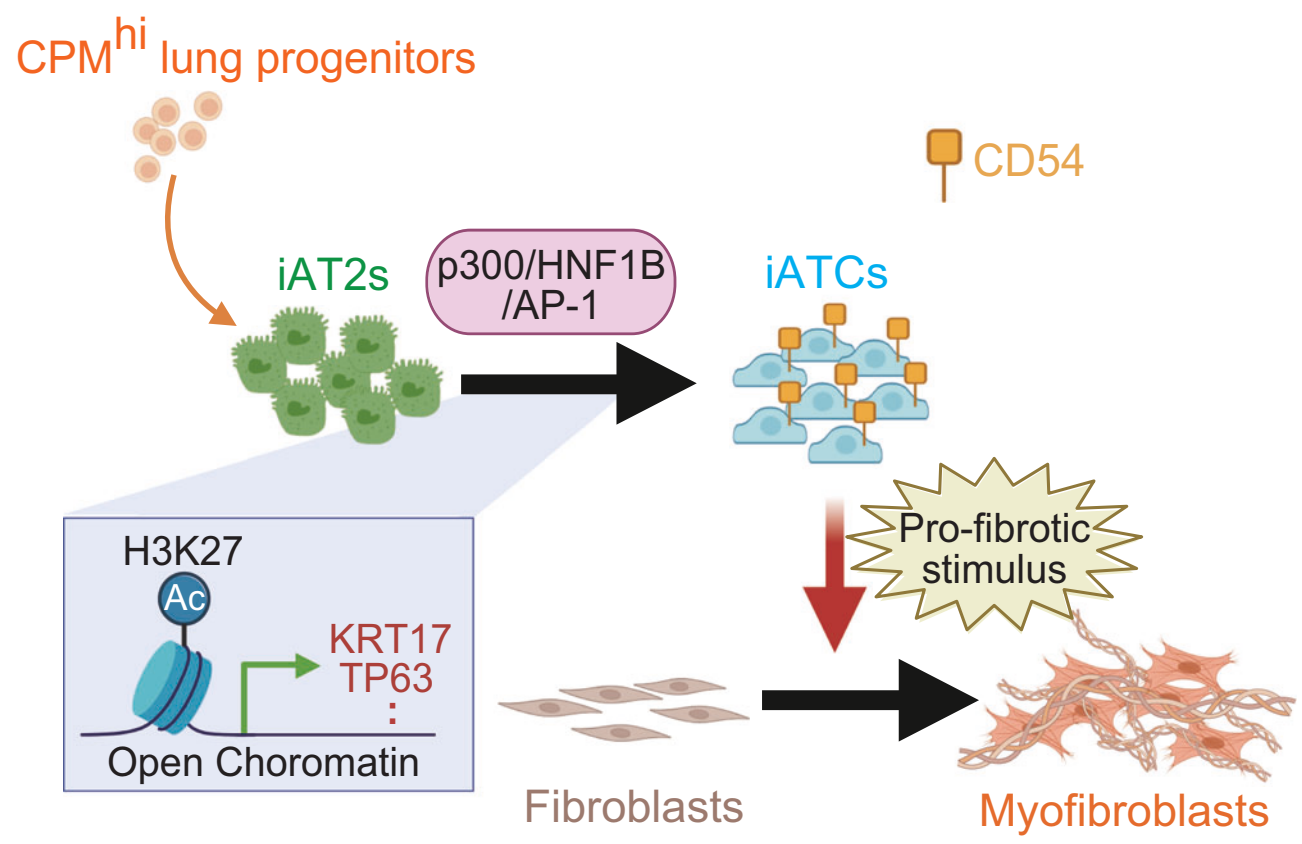


**Fig. 10** | Working model of p300-mediated regulation of iATCs differentiation and epithelial–fibroblast crosstalk.

based on shared barcodes between the fragment and matrix files. A ChromatinAssay object was created using the CreateChromatinAssay function. Gene annotation for chromatin accessibility peaks was performed using the Ensembl gene annotation via the EnsDb.Hsapiens.v86 package. To integrate scATAC-seq data with scRNA-seq data, label transfer was performed using the FindTransferAnchors and TransferData functions. Transfer anchors were identified between the scRNA-seq dataset and the scATAC-seq dataset using a canonical correlation analysis (CCA)-based approach. The transferred cell type labels were added to the scATAC-seq metadata. Latent semantic indexing (LSI) was performed for dimensionality reduction, followed by clustering using the FindClusters function. UMAP visualization was performed on the LSI-reduced space using the RunUMAP function. BED files were generated from the identified peak regions from each cluster. For visualization purposes, the files were transformed into bedgraph format using bedtools (v2.31)[73] and subsequently loaded into the Integrative Genomics Viewer. To identify differentially accessible chromatin regions between cell populations, we performed a logistic regression-based differential peak analysis using the FindMarkers function in Seurat with test.use = "LR". To obtain biologically relevant differential peaks, we applied the following criteria: present in at least 10% of cells in one of the compared groups (min.pct = 0.1), statistically significant with an adjusted *p*-value < 0.05, and showing a substantial increase in accessibility with log2FC > 0.5.

### BULK CUT&Tag

We prepared the CUT&Tag libraries according to previous literature[74,75], with slight modifications. The recombinant Protein A-conjugated Tn5 transposase (pA-Tn5) was extracted and purified from bacterial cell lysates of T7 Express lysY/Iq Competent *E. coli* (NEB, C3013) harboring 3XFlag-pA-Tn5-Fl plasmid (Addgene, 124601, a gift from Prof. Steven Henikoff) using a column filled with chitin beads (NEB, S6651L) after solubilization with sonication using Bioruptor II TYPE24 (Sonicbio, BR2024A). The transposase was assembled with an equimolar amount of the two types of oligo DNA adaptors (5′-TCGTCGGCAGCGTCAGATGTGTATAAGAGACAG-3′ and 5′-GTCTCGTGGGCTCGGAGATGTGTATAAGAGACAG-3′), each of which was pre-annealed with 5′-[PHO]CTGTCTCTTATACACATCT-3′, resulting in the pA-Tn5 transposome used in the following CUT&Tag library preparation. To prepare concanavalin A-conjugated magnetic beads, we first washed 500 μl of Dynabeads MyOne Streptavidin T1 (Thermo Fisher Scientific, DB65601) with PBS (pH 6.9) supplemented with 1 mM $CaCl_2$ and 1 mM $MnCl_2$ thrice. Then, we resuspended the beads with 500 μl PBS (pH 6.9) supplemented with 1 mM $CaCl_2$, 1 mM $MnCl_2$, and 0.01% Tween-20. We added 500 μg of biotin-conjugated concanavalin A (Sigma Aldrich, C2272-2MG) resuspended in 250 μl of PBS (pH 6.9) supplemented with 1 mM $CaCl_2$ and 1 mM $MnCl_2$ to the magnetic beads suspension. After incubation on a rotator at room temperature, we removed the supernatant after magnetic separation of the beads and then resuspended the beads in 500 μl PBS (pH 6.9) supplemented with 1 mM $CaCl_2$, 1 mM $MnCl_2$, and 0.01% Tween-20. Immediately before mixing the beads with the cells, we replaced the buffer with binding buffer (20 mM HEPES pH 7.9, 10 mM KCl, 1 mM $CaCl_2$, and 1 mM $MnCl_2$). To make the CUT&Tag libraries, the cultured cells were incubated in Accutase at 37 °C for 20 min, gently detached, washed with PBS supplemented with 1X Protease Inhibitor (Roche, 11873580001), and subsequently washed with wash buffer (20 mM HEPES pH 7.5, 150 mM NaCl, 0.25 mM spermidine, and 1X Protease Inhibitor). Then, we resuspended 50,000 cells in 500 μl of the wash buffer supplemented with 5 mM sodium butyrate. We added 2 μl of the concanavalin A-conjugated magnetic beads to the cell suspension. Next, after magnetic separation of the beads and removal of the supernatant, we resuspended the beads in 50 μL of ice-cold wash buffer, supplemented with 5 mM sodium butyrate, 0.05% digitonin, 2 mM EDTA, and 0.1% BSA together with 0.5 μl primary antibody against H3K27ac (MAB Institute, MABI0309), H3K4me3 (MAB Institute, MABI0304), H3K27me3 (MAB Institute, MABI0323), H3 (MAB Institute, MABI0301) or 1 μl primary antibody against p300 (Cell Signaling technology, #57625). After rotating incubation at room temperature for 3 h, the buffer containing the primary antibody was replaced with 100 μl of ice-cold wash buffer, supplemented with 5 mM sodium butyrate, 0.05% digitonin, together with 1 μl secondary rabbit anti-mouse IgG antibody (Abcam, ab46540), and was incubated at 4 °C for overnight. After washing the beads with Dig-300 buffer (20 mM HEPES pH 7.5, 300 mM NaCl, 0.5 mM spermidine, 0.01% digitonin, 1X Protease Inhibitor, and 5 mM sodium butyrate) twice, we applied 100 μl of the Dig-300 buffer containing 3.6 femtomol pA-Tn5 transposome assembled above and incubated the mixture for 1 h at room temperature. After washing the beads with the Dig-300 buffer twice, we resuspended the beads in 50 μl of the Dig-300 buffer supplemented with 10 mM $MgCl_2$ and incubated them at 37 °C for 1 h. Then, we removed the supernatant and resuspended the beads in 20 μl of 10 mM TAPS buffer (pH 8.5) together with 0.5 μl thermolabile proteinase K (NEB, P8111S) and incubated the mixture at 37 °C for 30 min, followed by incubation at 55 °C for 20 min to inactivate the proteinase. We directly performed PCR-amplification

of the library from the tube after adding the reaction mixture (KAPA BIOSYSTEMS, KK2602) using the primers listed in Supplementary Data 4. The amplified libraries were purified with 1.3 volumes of AMPure XP beads. For the p300 CUT&Tag experiments, the same reagent concentrations mentioned above were used, but both the total volumes of all solutions and the number of cells were doubled (100,000 cells per reaction). The libraries were sequenced with NovaSeq 6000 Sequencing System (Illumina) using NovaSeq 6000 SP Reagent Kit v1.5 (100 Cycles) (Illumina, 20028401) with the paired-end mode (61 bp each for Read1 and Read2). We used "DNA-mapping" in snakePipes (v2.7.3)[76] to trim adapter sequences with fastp (v0.23.2)[77] and to subsequently map the remaining sequences with bowtie2 (v2.4.5)[78] against the reference sequence created from the human T2T-CHM13v2.0 genome assembly without Y chromosome together with the complete genome sequence of Escherichia coli strain BL21 (GenBank: CP010816.1) and the 3XFlag-pA-Tn5-Fl plasmid sequence (Addgene, 124601). Then, we obtained a bam file only with properly-paired alignments mapped to the human chromosomes 1 to 22 and X with the MAPQ score greater than or equal to 10 using samtools view command (v1.15.1)[79] with the option "-f 2 -F 256 -x XA -q 10." We further created bigwig tracks from the bam files using "bamCoverage" in deeptools (v3.5.1)[80] with the option "--binSize 25 --effectiveGenomeSize 3054815472 --normalizeUsing CPM." We called peaks with MACS2 callpeak (v2.2.7.1)[81] with the option "-f BAM -g hs --nomodel" from the aligned reads for H3K27Ac, H3K4me3, and H3K27me3 taking the alignment from the anti-H3 CUT&Tag from the same sample as the control, using the snakemake wrapper v2.1.0 in a snakemake workflow[82]. For p300, peaks were called without a control sample under the same parameters. For visualization, bigwig files generated from biological duplicates were merged using bigWigMerge, resulting in a single file. Significantly increased peaks were identified using DiffBind (ver. 3.8.4). The resulting BED file was generated, and its coordinates were lifted over from T2T-CHM13v2.0 to the hg38 genome assembly for motif analysis.

### Motif analysis
BED files were used as input for HOMER (v4.11)[83] to perform motif enrichment analysis. Default settings of HOMER's findMotifsGenome.pl commands were applied using the hg38 reference genome. The analysis identified transcription factor motifs enriched in specific peaks, which were further used to interpret regulatory elements associated with each cluster. Motif-binding sites for each transcription factor were identified using HOMER's annotatePeaks.pl function with the corresponding motif files, allowing precise localization of motif occurrences within the enriched peaks. Genes associated with these motif-binding regions were then predicted using GREAT (v4.0.4)[51].

### siRNA transfection
CPM$^{hi}$ lung epithelial progenitor cells ($1 \times 10^5$ cells) were seeded onto micro-patterned culture plates and transfected simultaneously with 10 nM siRNAs using the Lipofectamine RNAiMAX reagent (Thermo Fisher Scientific, 13778150), according to the manufacturer's instructions. The transfection mixture was prepared by combining 1 μL of Lipofectamine RNAiMAX with 50 μL of Opti-MEM (Thermo Fisher Scientific) per well. The siRNAs used were those targeting human ATF3 (Thermo Fisher Scientific, s1699) or HNF1B (Dharmacon, L-009721-00-0005), along with a non-targeting negative control siRNA (Dharmacon, D-001810-10).

### Animals
Six-week-old male C57BL/6JJcl mice were obtained from CLEA Japan, Inc. All experiments involving animal use were performed per the guide for the care and use of laboratory animals of the Kyoto University, and the procedures were approved by the institutional committee of the Kyoto University (22-180-5).

### Mouse model for BLM-induced pulmonary fibrosis
For BLM administration, 7-week-old male mice were anesthetized with isoflurane (Viatris, 2155944), followed by oropharyngeal aspiration (OA) of BLM (1.75 mg/kg) in 100 μL of saline. Control groups received saline alone. Mice were administered either saline (control group) or CBP30 (10 mg/kg, O.A.) on days 2, 4, and 6. Histological and qPCR analyses were performed on day 7.

### Statistical analysis
Data are presented as mean ± standard error of the mean (SEM). The number of biological replicates and statistical tests are described in each figure legend. All statistical tests were performed using Prism 10 software (GraphPad). A $p$-value of <0.05 was considered statistically significant.

### Reporting summary
Further information on research design is available in the Nature Portfolio Reporting Summary linked to this article.

## Data availability
The accession numbers for bulk RNA-seq, bulk CUT & Tag, single-cell RNA-seq, and single-cell ATAC-seq data reported in this study are GSE289676, GSE289678, GSE289679, GSE289682, GSE289683, GSE291333, GSE289846, GSE290014, [PRJDB37980], [PRJDB37982], and [PRJDB37983], respectively. The proteomic analysis data was deposited to the ProteomeXchange Consortium via jPOSTrepo [https://repository.jpostdb.org/] with the dataset identifier JPST003570 (PXD060050). Source data are provided with this paper.

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

## Acknowledgements

We thank all of the members of the Gotoh lab (CiRA, Kyoto University); Dr. Yuko Ohnishi, Dr. Michiaki Nagasawa, Dr. Kazuhisa Nakao, Dr. Naoki Ohmae, Dr. Shuhei Kanagaki, and Dr. Keita Moriguchi (Kyorin Pharmaceutical); Dr. Ryuta Mikawa, Dr. Akiko Nakano-Kobayashi, Dr. Atsuyasu Sato, Dr. Toyohiro Hirai, Dr. Masatoshi Hagiwara (Graduate School of Medicine, Kyoto University), Dr. Mitsuru Morimoto (RIKEN BDR), Dr. Takuji Suzuki (Graduate School of Medicine, Chiba University) for helpful discussion and comments; Ms. Kazusa Okita, Ms. Kazumi Deguchi, and Ms. Satoko Sakurai (CiRA, Kyoto University) for technical assistance with the RNA-seq and scRNA-seq experiments; Ms. Yuka Motohiro (ASHBi, Kyoto University) for technical assistance with the scATAC-seq and CUT&Tag experiments; Ms. Megumi Narita (CiRA, Kyoto University) for technical assistance with the proteomic analysis; all of the members of the Medical Research Support Center, Kyoto University for adjustment of laboratory equipment; Single-cell Genome Information Analysis Core (SignAC) at WPI-ASHBi, Kyoto University for their support; Tosoh Ltd. for providing the micro-patterned culture plates; MGI Tech Co., Ltd. for providing the scATAC-seq preparation kit. Some Figures were created in BioRender.com. This study was supported by the Collaboration Research Fund to Kyoto University from Kyorin Pharmaceutical Co. Ltd., the iPS Cell Research Fund for CiRA at Kyoto University, and AMED (JP23bm1323001, JP23bm1423004, JP25bk0104190, JP24gm1910009) (to S.G.).

## Author contributions

Conceptualization: Y.T., S.K., and S.G.; methodology: Y.T., and A.M.; validation: Y.T.; formal analysis: Y.T., T.T., M.I., and T.Y.; investigation: Y.T, and A.M.; resources: Y.T., and S.G.; data curation: Y.T., T.T., M.I., and T.Y.; writing – original draft: Y.T., S.K., T.T., M.I., and S.G.; writing – review & editing: Y.T., S.K., and S.G.; visualization: Y.T.; supervision: S.G.; project administration: S.G.

## Competing interests

Y.T. and A.M. are employees and shareholders of Kyorin Pharmaceutical Co., Ltd. S.G. is a founder and shareholder of HiLung. M.I. is a scientific adviser for xFOREST Therapeutics without a salary, and all other authors declare no competing interests. S.G. filed the patents related to the methods of generating iPSC-derived lung cells in this study: WO2014168264A1, WO2016143803A1, and WO2016148307A1.
