## [Transparent Peer Review file · Nature Communications]

Human iPSC-based Modeling of Pulmonary Fibrosis Reveals p300/CBP Inhibition Suppresses Alveolar Transitional Cell State

Corresponding Author: Dr Shimpei Gotoh

Version 0:

Reviewer comments:

Reviewer #1

(Remarks to the Author)

Tsutsui and colleagues present a compelling dataset addressing the generation of a critical cell type in lung disease. They have developed well-designed in vitro model systems that recapitulate features of human lung disease. This approach is commendable as it circumvents some of the limitations inherent in rodent models and enables functional screens, such as the one central to this manuscript. Given the ongoing debate surrounding the AT2-to-AT1 transitional population—referred to by many names—and its role in disease pathogenesis, this study represents a timely and impactful contribution to the lung biology field. The authors should be commended for the rigor and scope of their work.

Major Comments

1. A central thesis of the manuscript is that p300/CBP inhibition suppresses the emergence of ATCS cells from AT2 cells, as depicted in the first part of the schematic in Figure 6. A reduction in iATC markers is also shown in Figure 5a. The primary mechanistic data on p300/CBP come from Figure 6, where the authors state that H3K27ac is a direct target of p300/CBP inhibition. In the supplement, they show that H3K27ac is reduced following CBP30 treatment. However, the key mechanistic experiment—CUT&Tag—is performed only in untreated cells. These data are somewhat expected, as H3K27ac is localized where anticipated across the three conditions. The authors propose that p300 regulates “key ATCS” transcription factors, but the evidence is indirect and based on H3K27ac localization. A critical experiment that would strengthen the manuscript is to perform H3K27ac CUT&Tag in iATCs cultured in PAL+ with and without p300/CBP inhibitor. Comparing the differential peaks in the presence of the drug would more directly support the authors’ mechanistic claims.

2. There is a missed opportunity to explore the fate potential of ATCS cells using the authors’ model system. The manuscript shows that iAT2 cells can self-renew, differentiate into iAT1 cells, or transition into iATCs. The introduction rightly emphasizes a central unanswered question in the field: do transitional cells accumulate because they are terminally differentiated, or because the environment fails to support further differentiation? This is difficult to assess in vivo, but the authors’ model system is well-suited to address it. A key experiment would be to isolate iATCs (such as with ICAM) and culture them under the same three conditions used for iAT2 cells: DCKI+3i, DCI+LATSi, and PAL+. This would determine whether iATCs can revert to AT2 cells, differentiate into AT1 cells, or self-renew. This experiment would not be a mere curiosity—it would significantly advance understanding of transitional cell biology and strengthen the manuscript’s impact.

Minor Comments

- Consider briefly summarizing the various names used for ATCS cells in the literature to provide context. While I like the name, this will help orient readers.
- Define “morphological cytotoxicity.”
- In Figure 2, clearly label which cell types are analyzed (e.g., fibroblasts vs. epithelium). Also indicate the directionality (up- vs. down-regulation) in the pathway analysis.
- In Figure 2d, clarify why the authors switch from BML to TGF- β 1 to induce fibrosis. Does bleomycin have no effect on HFLF alone? If so, that’s an interesting finding worth noting.
- Staining for p300 should be nuclear; however, in several figures it appears to overlap with SFN, which is cytoplasmic. Higher-resolution images are needed to resolve this discrepancy.
- In Figure 4b, the color scheme is difficult to distinguish due to similar shades—consider adjusting to improve clarity.
- Figure 5c: Include a negative control (non-iATC organoids) to show ICAM does not label all cells nonspecifically.

- Figure 5h: Add a brief explanation for changing fibroblast types from NHLF to fetal.
- Figure 5j: Clarify what population is used for iATC(-). Is it NHLF alone, or co-cultured with CD54-negative cells?
- In Figure 6, explain the purpose of adding Hoechst to the sorting protocol.
- For the CUT&Tag experiments on iAT2 and iAT1 cells, note that these are heterogeneous populations (as shown by your scRNA-seq data) and were not sorted specifically. The conclusions remain valid, but clarifying this avoids misinterpretation.
- Consider using scRNA-seq data with tools like CellChat or NicheNet to identify potential mediators of epithelial-fibroblast communication. This would strengthen the epithelial-mediated fibrosis hypothesis.
- For space, consider moving the fibroblast-only gel contraction experiment and Figure 4i to the supplementary material.
- For readers less familiar with the group's iPSC models, briefly describe how iAT2 cells are generated and note that similar approaches have been validated by other groups.

Reviewer #2

(Remarks to the Author)

Reviewer #3

(Remarks to the Author)

In this manuscript, Tsutsui et al. describe a series of experiments in iPSC-derived lung organoids focused around mechanisms of pulmonary fibrosis. The problem is important and the group is a leader in this area, so I read the manuscript with significant interest. Overall I think the work here is well done and interesting. The manuscript is well written and the figures are clear. There is some significant technical advancement present in the work. These aspects make me enthusiastic. However, there are two somewhat distinct questions addressed, and they are currently relatively distinct, giving the impression of two short manuscripts (both good but somewhat incomplete) combined into a single longer and somewhat less focused body of work. I would recommend one (large) experiment to the authors to address this concern. The authors should strongly consider evaluating p300 binding in their Cut and Tag assay with and without bleomycin, ideally with temporality. This would define the putative TFs which co-bind with p300 with and without fibrotic stimuli, allowing some insight into the mechanism by which the p300 inhibition prevents fibrosis and therefore transition to pro-fibrotic stressed cell states, which would harmonize the two pieces of work and be of significant therapeutic importance. Presenting this work in a revised final figure would make this manuscript of high significance in the field.

Version 1:

Reviewer comments:

Reviewer #1

(Remarks to the Author)

We think that the authors responded extremely well to our considerations and requests. The new mechanistic data, referring to the CUT&RUN, as well as the new differentiation capacity data bolster the conclusions.

We have no additional comments to make on the experiments and data interpretations.

Very wonderful job!

Reviewer #2

(Remarks to the Author)

Reviewer #3

(Remarks to the Author)

The authors have addressed my concerns with a nice body of new data, and in my view have addressed the concerns of the other reviewer as well. I am supportive of publication at this time.

A point-by-point response to the Reviewers' comments:

Reviewer #1-2

Tsutsui and colleagues present a compelling dataset addressing the generation of a critical cell type in lung disease. They have developed well-designed in vitro model systems that recapitulate features of human lung disease. This approach is commendable as it circumvents some of the limitations inherent in rodent models and enables functional screens, such as the one central to this manuscript. Given the ongoing debate surrounding the AT2-to-AT1 transitional population—referred to by many names—and its role in disease pathogenesis, this study represents a timely and impactful contribution to the lung biology field. The authors should be commended for the rigor and scope of their work.

We sincerely thank the reviewer for the valuable comments. The manuscript has been revised accordingly, and our point-by-point responses are provided below.

Major comments

1) A central thesis of the manuscript is that p300/CBP inhibition suppresses the emergence of ATCS cells from AT2 cells, as depicted in the first part of the schematic in Figure 6. A reduction in iATC markers is also shown in Figure 5a. The primary mechanistic data on p300/CBP come from Figure 6, where the authors state that H3K27ac is a direct target of p300/CBP inhibition. In the supplement, they show that H3K27ac is reduced following CBP30 treatment. However, the key mechanistic experiment—CUT&Tag—is performed only in untreated cells. These data are somewhat expected, as H3K27ac is localized where anticipated across the three conditions. The authors propose that p300 regulates “key ATCS” transcription factors, but the evidence is indirect and based on H3K27ac localization. A critical experiment that would strengthen the manuscript is to perform H3K27ac CUT&Tag in iATCs cultured in PAL+ with and without p300/CBP inhibitor. Comparing the differential peaks in the presence of the drug would more directly support the authors' mechanistic claims.

Response to Major comment 1:

We appreciate the reviewer's insightful suggestion regarding the need to assess the effects of p300/CBP inhibition on H3K27ac distribution in iATCs. In response, we performed H3K27ac CUT&Tag in iATCs cultured under PAL+ conditions, with or without p300/CBP inhibitors (CBP30 and GNE781), to directly evaluate the epigenetic consequences of p300/CBP inhibition (Fig. 8a). As p300/CBP inhibition reduced the number of CD54⁺ cells, we first optimized an alternative cell isolation method for CUT&Tag using brief Hoechst staining, which successfully enriched peripheral cells corresponding to CD54⁺ iATCs (Supplementary Fig. 11a–b). As expected, global H3K27ac levels were reduced following

inhibitor treatment. Differential peak analysis using DiffBind revealed that H3K27ac peaks were significantly decreased in enhancer- and promoter-associated regions, consistent with the established role of p300/CBP in histone acetylation (Supplementary Fig. 12a). Motif analysis of the decreased H3K27ac peaks showed enrichment of AP-1 motifs, consistent with our previous comparison of CD54⁺ iATCs with iAT2s/iAT1s-enriched cells (Figs. 7c and 8b–c; Supplementary Fig. 12b). Notably, the HNF1B motif ranked among the top motifs in the p300/CBP inhibitor–decreased H3K27ac peaks, following broadly detected motifs such as TEAD and FOX, which appeared across multiple alveolar epithelial lineages. In contrast, HNF1B was not highly ranked in iAT1- or iAT2-enriched cells, suggesting a potential HNF1B-specific role in the regulation of iATCs by p300/CBP (Figs. 7c and 8b–c; Supplementary Fig. 12b). Consistently, motif analysis of p300-bound regions also revealed enrichment of these transcription factors, reinforcing their potential roles in iATCs differentiation (Supplementary Fig. 12c). To further define the downstream regulatory programs of p300/CBP, GREAT analysis of H3K27ac peaks decreased by p300/CBP inhibition identified AP-1– and HNF1B-associated target genes enriched in epithelial stress and fibrogenic pathways, including the RHO GTPase cycle, FAK signaling, and RAF/MAPK cascade (Fig. 8d–f). These results suggest that p300/CBP regulates iATCs differentiation through AP-1- and HNF1B-dependent transcriptional control. We also conducted functional inhibition experiments targeting AP-1 and HNF1B, which collectively support the involvement of these transcription factors in regulating ATCS differentiation and provide new insights into p300/CBP-mediated control of ATCS via H3K27ac (Fig. 9e–g, Supplementary Fig. 15a). Taken together, these new results demonstrate that p300/CBP inhibition reduces H3K27ac at key regulatory regions and modulates transcription factors critical for ATCS differentiation, thereby reinforcing our mechanistic model. The corresponding results have been incorporated into the revised manuscript (page 12, line 324 ~ page 13, line 356).

2) There is a missed opportunity to explore the fate potential of ATCS cells using the authors' model system. The manuscript shows that iAT2 cells can self-renew, differentiate into iAT1 cells, or transition into iATCs. The introduction rightly emphasizes a central unanswered question in the field: do transitional cells accumulate because they are terminally differentiated, or because the environment fails to support further differentiation? This is difficult to assess in vivo, but the authors' model system is well-suited to address it. A key experiment would be to isolate iATCs (such as with ICAM) and culture them under the same three conditions used for iAT2 cells: DCKI+3i, DCI+LATSi, and PAL+. This would determine whether iATCs can revert to AT2 cells, differentiate into AT1 cells, or self-renew. This experiment would not be a mere curiosity—it would significantly advance understanding of transitional cell biology and strengthen the manuscript's impact.

Response to Major comment 2:

We thank the reviewer for this thoughtful and constructive suggestion regarding the need to assess the fate potential of ATCS cells. In response, we conducted additional experiments using our previously established SFTPC^{GFP} AGER^{mCherry-HiBiT} dual-reporter iPS cell line combined with the on-gel alveolar epithelial spheroid culture system (Ohnishi et al., *Stem Cell Reports*, 2024; ref. 40.) to directly evaluate the differentiation potential of CD54⁺ iATCs. These experiments demonstrated that CD54⁺ iATCs lacked self-renewal capacity (Supplementary Fig. 7a–b) but were able to revert to functional iAT2s when cultured under DCIK+3i conditions (Fig. 6a–d; Supplementary Fig. 7c–e) and could also differentiate directly into iAT1s when exposed to DCI+LATSi medium (Fig. 6e–i).

These findings provide direct experimental evidence for the bidirectional differentiation potential of iATCs and clarify their position within the alveolar epithelial lineage hierarchy. The corresponding results have been incorporated into the revised manuscript (page 10, line 275 ~ page 11, line 298). Collectively, these findings further suggest that the accumulation of transitional cells in disease contexts may reflect an impaired differentiation environment rather than a terminally differentiated fate.

Minor comments

1) Consider briefly summarizing the various names used for ATCS cells in the literature to provide context. While I like the name, this will help orient readers.

Response to Minor comment 1:

We thank the reviewer for this helpful suggestion. As recommended, we have briefly summarized the various terminologies used for ATCS in the literature to provide context. Specifically, we noted that in mice, these cells have been described as KRT8⁺ ADI, PATS, or DATPs (page 3, line 51~52), whereas in humans, they are referred to as *KRT5*⁻/*KRT17*⁺ cells, aberrant basaloid cells, or PATS-like cells (page 3, line 56~57).

2) Define “morphological cytotoxicity.”

Response to Minor comment 2:

We appreciate the reviewer’s request for clarification. To define “morphological cytotoxicity,” we conducted additional analyses examining the relationship between organoid morphology and cytotoxic responses. Organoids exhibiting darkened epithelial spheroids—previously categorized as showing morphological cytotoxicity—displayed increased expression of

NOXA, an apoptosis-related gene, indicating that their observed loss of contraction resulted from compound-induced cell death (Supplementary Fig. 1b–c). Based on these findings, such organoids were considered as false positives and excluded from subsequent analyses. We have included this clarification and supporting data in the revised manuscript (page 5, line 103 ~ 106), adding the following description:

“Organoids exhibiting darkened epithelial spheroids displayed increased expression of the apoptosis-related gene *NOXA*, indicating that their loss of contraction resulted from compound-induced cell death (Supplementary Fig. 1b, c). Therefore, these organoids were considered false positives and excluded from further analysis.”

3) In Figure 2, clearly label which cell types are analyzed (e.g., fibroblasts vs. epithelium). Also indicate the directionality (up- vs. down-regulation) in the pathway analysis.

Response to Minor comment 3:

We thank the reviewer for this helpful suggestion. Figure 2b–d and the corresponding Supplementary Fig. 2a–b and f have been revised to clearly label the analyzed cell types and indicate the directionality (up- or down-regulation) in the pathway analysis.

4) In Figure 2d, clarify why the authors switch from BML to TGF-β1 to induce fibrosis. Does bleomycin have no effect on HFLF alone? If so, that's an interesting finding worth noting.

Response to Minor comment 4:

We thank the reviewer for raising this important point. To clarify, we have added the following explanation in the revised *Results* section (page 6, line 128 ~ 132):

“We next examined which cell types mediated the anti-contraction effects of the p300/CBP inhibitors. Our previous findings demonstrated that TGF-β1 stimulation induced gel contraction in a 3D fibroblast-only culture system, whereas BLM stimulation did not.³⁰ This indicates that the BLM-induced contraction observed in FD-AOs is mediated by epithelial cells rather than being a direct effect on fibroblasts.”

To support this clarification, we have included below an image from our previous study (Suezawa et al., Stem Cell Reports, 2021; ref. 30, figure. 1), demonstrating that bleomycin did not induce contraction in a fibroblast-only gel.

5) Staining for p300 should be nuclear; however, in several figures it appears to overlap with SFN, which is cytoplasmic. Higher-resolution images are needed to resolve this discrepancy.

Response to Minor comment 5:

We thank the reviewer for this valuable comment and apologize for any confusion caused by the images in the original submission. We have replaced the representative images with clearer ones, wherein the nuclear localization of Act-p300 is indicated by arrowheads (Fig. 2e and Fig. 3d). For clarity, the updated images are also provided below in this response

Fig. 2e
(Organoids)

Fig. 3d
(Mice)

6) *In Figure 4b, the color scheme is difficult to distinguish due to similar shades—consider adjusting to improve clarity.*

Response to Minor comment 6:

We thank the reviewer for the suggestion. Figure 4b has been revised with an improved color scheme for clarity.

7) *Figure 5c: Include a negative control (non-iATC organoids) to show ICAM does not label all cells nonspecifically.*

Response to Minor comment 7:

We thank the reviewer for this helpful suggestion. In the revised manuscript, the immunofluorescence data have been updated to include organoids cultured under iAT2- and iAT1-inducing conditions as negative controls (Fig. 5c). CD54 expression was barely detectable under iAT2-inducing conditions and weakly observed under iAT1-inducing conditions, confirming that CD54 labeling is not nonspecific. Accordingly, we have added the following description to the *Results* section (page 9, lines 247 ~ page 10, lines 253): “Immunofluorescence analysis revealed that CD54 expression was barely detectable under iAT2-inducing conditions and was weakly observed under iAT1-inducing conditions (Fig. 5c). Conversely, CD54 was prominently expressed and colocalized with ATCS markers under iATCs-inducing conditions (Fig. 5c; Supplementary Fig. 5c). These results suggest that CD54 serve as a potential surface antigen for isolating iATCs, consistent with its gene expression levels (Fig. 5b; Supplementary Fig. 5d).”

8) *Figure 5h: Add a brief explanation for changing fibroblast types from NHLF to fetal.*

Response to Minor comment 8:

We thank the reviewer for this constructive suggestion. In the revised manuscript, we have added the following explanation to clarify the rationale for using NHLFs instead of fetal lung fibroblasts (page 10, lines 255 ~ 260).

“We previously demonstrated that primary human fetal lung fibroblasts supported the induction and maintenance of iAT2s, unlike primary normal human adult lung fibroblasts (NHLFs) that cannot induce iAT2 differentiation.²⁹ As idiopathic pulmonary fibrosis—a major

form of pulmonary fibrosis—predominantly develops in adulthood, we investigated whether co-culture of NHLFs with CD54⁺ iATCs promotes fibroblast activation (Fig. 5h).”

9) Figure 5j: Clarify what population is used for iATC(-). Is it NHLF alone, or co-cultured with CD54-negative cells?

Response to Minor comment 9:

We thank the reviewer for this helpful comment. The iATCs (-) condition represents NHLF alone, not co-cultured with CD54-negative cells. To avoid confusion, we have clarified this in the revised figure 5j by labeling the condition as “alone.”

10) In Figure 6, explain the purpose of adding Hoechst to the sorting protocol.

Response to Minor comment 10:

We thank the reviewer for this insightful comment. In our previous study (Masui et al., *Stem Cell Reports*, 2024; ref. 37.), we demonstrated that iAT2s at the periphery of colonies generated by the micropatterned culture system can be isolated and enriched using short-term Hoechst staining (30 min). In the present study, we additionally confirmed that iAT1s are also predominantly localized at the colony periphery and can be enriched using the same Hoechst-based approach. In response to the reviewer’s suggestion, these new details for iAT1s have been included in the revised manuscript (Supplementary Fig. 8a–b). By comparing these Hoechst-enriched iAT1s (iAT1-enriched cells) and previously characterized iAT2s (iAT2-enriched cells) with CD54⁺ iATCs in histone CUT&Tag analysis, we aimed to elucidate the epigenetic dynamics underlying alveolar epithelial differentiation (Fig. 7a; Supplementary Fig. 8c). We have added the following sentences to the revised manuscript accordingly (page 11, lines 303 ~ 310):

“In our previous study, we demonstrated that iAT2s at the periphery of colonies induced by the micropatterned culture system can be isolated and enriched using short-term Hoechst staining (30 min)³⁷. Similarly, iAT1s are predominantly localized at the colony periphery and can be enriched using the same approach (Supplementary Fig. 8a–b). By comparing these enriched populations of iAT2s (iAT2-enriched cells) and iAT1s (iAT1-enriched cells) with CD54⁺ iATCs using CUT&Tag profiling of histone modifications, we elucidated the epigenetic landscapes and associated transcription factors that shape the identity and persistence of iATCs (Fig. 7a; Supplementary Fig 8c).”

11) For the CUT&Tag experiments on iAT2 and iAT1 cells, note that these are heterogeneous populations (as shown by your scRNA-seq data) and were not sorted specifically. The conclusions remain valid, but clarifying this avoids misinterpretation.

Response to Minor comment 11:

We appreciate the reviewer's valuable comment and apologize for the potentially misleading description in the original version. To clarify this point, we have revised the terminology to "iAT2-enriched cells" and "iAT1-enriched cells," which more accurately represent the heterogeneity within these populations. Accordingly, these terms have been updated in the figure labels (Fig. 7; Supplementary Fig. 9) and in the *Results* section (page 11, lines 307 ~ 308) to avoid any potential misinterpretation regarding the cell populations used for the CUT&Tag experiments.

12) Consider using scRNA-seq data with tools like CellChat or NicheNet to identify potential mediators of epithelial-fibroblast communication. This would strengthen the epithelial-mediated fibrosis hypothesis.

Response to Minor comment 12:

We thank the reviewer for this insightful suggestion. While our scRNA-seq dataset focuses on epithelial cells derived from the micropatterned culture system, we additionally analyzed bulk RNA-seq data from normal human lung fibroblasts (NHLFs) co-cultured with or without CD54⁺ iATCs to investigate potential epithelial–fibroblast interactions (Supplementary Fig. 6a). Using these data, we identified receptor genes that were upregulated in fibroblasts upon co-culture with CD54⁺ iATCs (Supplementary Fig. 6a). To explore their potential epithelial ligands, we referred to the human CellChat database (Jin S et al., *Nat Commun*, 2021; ref. 48.) and examined the expression of the corresponding ligand genes in CD54⁺ iATCs using our scRNA-seq dataset (Supplementary Fig. 6c). This analysis suggested that TGFβ, WNT, and SEMA signaling pathways may contribute to epithelial–fibroblast communication (Supplementary Fig. 6d). In the revised manuscript, we have added the following description (page 10, lines 268 ~ 273):

"NHLFs co-cultured with iATCs exhibited increased expression of several receptor genes, including *ITGAV*, *ITGB6*, *TGFBR1*, *FZD8*, and *NRP2* (Supplementary Fig. 6a). To identify potential mediators of epithelial–fibroblast communication, we referred to the human CellChat database⁴⁸, which indicated that TGFβ, WNT, and SEMA signaling pathways

represent possible ligand–receptor interactions between iATCs and fibroblasts based on their expression patterns (Supplementary Fig. 6c–d).”

13) For space, consider moving the fibroblast-only gel contraction experiment and Figure 4i to the supplementary material.

Response to Minor comment 13:

We thank the reviewer for the suggestion. As recommended, the fibroblast-only gel contraction experiment and Figure 4i have been moved to the supplementary material (Supplementary Figs. 2c–e and 4c).

14) For readers less familiar with the group’s iPSC models, briefly describe how iAT2 cells are generated and note that similar approaches have been validated by other groups.

Response to Minor comment 14:

We thank the reviewer for the suggestion. As recommended, we have added a brief description of how iAT2 cells are generated and noted that similar approaches have been validated by other groups. For clarity, a schematic illustration has been also included in the supplementary material (Supplementary Fig. 1a).

Reviewer #3

In this manuscript, Tsutsui et al. describe a series of experiments in iPSC-derived lung organoids focused around mechanisms of pulmonary fibrosis. The problem is important and the group is a leader in this area, so I read the manuscript with significant interest. Overall I think the work here is well done and interesting. The manuscript is well written and the figures are clear. There is some significant technical advancement present in the work. These aspects make me enthusiastic. However, there are two somewhat distinct questions addressed, and they are currently relatively distinct, giving the impression of two short manuscripts (both good but somewhat incomplete) combined into a single longer and somewhat less focused body of work. I would recommend one (large) experiment to the authors to address this concern. The authors should strongly consider evaluating p300 binding in their Cut and Tag assay with and without bleomycin, ideally with temporality. This would define the putative TFs which co-bind with p300 with and without fibrotic stimuli, allowing some insight into the mechanism by which the p300 inhibition prevents fibrosis and therefore transition to pro-fibrotic stressed cell states, which would harmonize the two pieces

of work and be of significant therapeutic importance. Presenting this work in a revised final figure would make this manuscript of high significance in the field.

Response to Reviewer #3's comment

We sincerely thank the reviewer for this valuable suggestion. To address this point, we have performed p300 CUT&Tag analysis in FD-AOs treated with or without bleomycin to characterize the dynamics of p300 binding during iATCs differentiation and the transition toward a pro-fibrotic state (Fig. 9a). At 3 days after bleomycin treatment (the time point when bleomycin was washed out and compounds were added), motif analysis revealed enrichment of several stress- and differentiation-related transcription factors in epithelial cells, including p53 and HNF1B (Fig. 9a; Supplementary Data 3). Notably, HNF1B was identified here for the first time, consistent with its enrichment in the iATCs CUT&Tag analysis (Figs. 7c and 8b–c; Supplementary Fig. 12b). In contrast, analysis at 6 days after bleomycin treatment (the time point for evaluating gel contraction) showed enrichment of SMAD2/3, HIF1A/1B, and NF- κ B (p50/p52) motifs (Fig. 9a; Supplementary Data 3). AP-1, TEAD, and FOX family motifs were consistently enriched at both time points, suggesting their persistent involvement in the maintenance and progression of the fibrotic epithelial phenotype. Conversely, only a few p300 peaks were significantly reduced after bleomycin treatment, and motif enrichment was limited (Supplementary Fig. 13c–d). Motif analysis of H3K27ac peaks decreased by bleomycin treatment revealed enrichment of NKX2.1 and FOXA2—key transcription factors essential for alveolar epithelial development and homeostasis—which were conversely enriched in peaks increased upon treatment with p300/CBP inhibitors (Supplementary Fig. 14c–f). These findings suggest that fibrotic stimulation suppresses the maintenance of alveolar epithelial lineage-defining transcription factors, thereby promoting aberrant differentiation and transition toward a pro-fibrotic epithelial state, whereas p300/CBP inhibition may restore proper transcriptional programs by reactivating these lineage-associated factors. Furthermore, to validate the involvement of AP-1 family transcription factors in our iPSC-derived model, we evaluated the effects of AP-1 inhibitors (T-5224 and SR11302). These inhibitors significantly suppressed iATCs differentiation and gel contraction in bleomycin-treated FD-AOs (Fig. 9c–d; Supplementary Fig. 15a). We have incorporated these new data and results into the revised manuscript (page 13, lines 358 ~ page 14, lines 397). These experiments define the dynamic behavior of p300 and its cooperating transcription factors under fibrotic stimulation, directly addressing the reviewer's insightful suggestion and substantially strengthening the mechanistic framework of our study. Importantly, these findings also provide a mechanistic rationale for targeting the p300/CBP–AP-1 axis as a potential therapeutic strategy to prevent the aberrant differentiation of alveolar epithelium that drives lung fibrosis.